# The transcription factor Dof3.6/OBP3 regulates iron homeostasis in Arabidopsis

Peipei Xu [1✉], Yilin Yang[1,2], Zhongtian Zhao[1,2], Jinbo Hu[1], Junyan Xie[1], Lihua Wang[1], Huiqiong Zheng [1✉] & Weiming Cai [1✉]

## Abstract

**Iron is an essential element for plants. Iron uptake by plants is highly regulated, but the underlying mechanism is poorly understood. Using a truncated fragment of the iron deficiency-responsive bHLH100 gene promoter, we screened the Arabidopsis transcription factor yeast one-hybrid (Y1H) library and identified the DOF family protein, OBP3, as a crucial component of the iron deficiency-signaling pathway. OBP3 is a transcriptional repressor with a C-terminal activation domain. Its expression is induced by iron deficiency. The transgenic lines that overexpress OBP3 exhibited iron overload and premature leaf necrosis, while the obp3 mutant was less tolerant of iron deficiency. It was discovered that OBP3 directly targets the Ib subgroup of bHLH gene promoters. OBP3 interacts with the bHLH transcription factor ILR3 (IAA-LEUCINE RESISTANT3), and their interaction enhances the DNA-binding ability and transcriptional promoting activity of OBP3, resulting in the positive regulation of iron deficiency-response genes. In addition, the E3 Ligase BRUTUS facilitates 26S proteasome-mediated degradation of OBP3 protein to prevent excessive iron uptake in plants. In conclusion, our research emphasizes the vital role of OBP3 in regulating plant iron homeostasis.**

**Keywords** DOF Transcription Factor; Iron Homeostasis; Subgroup Ib/IVc *bHLH* Transcription Factor; Y1H Screening; Y2H Screening
**Subject Category** Plant Biology

## Introduction

Iron (Fe) is essential for almost all metabolic processes in plants and is an irreplaceable cofactor in hundreds of enzymes, including photosynthesis, DNA replication, nitrate assimilation and nitrogen fixation, respiration and reactive oxygen species (ROS) defense (Kermeur et al, 2023; Liang, 2022; Velez-Bermudez and Schmidt, 2023). However, excessive iron accumulation in plants can lead to the Fenton reaction, converting hydrogen peroxide to another ROS,

the hydroxyl radical, which is toxic to plants, and therefore plants must strictly regulate iron homeostasis (Li et al, 2023). Although soil iron content in most areas is very high, iron is mainly in the form of ferric ($Fe^{3+}$) oxyhydrates, with very low iron oxide solubility. Increased bioavailability of iron is achieved by a series of reduction and chelation mechanisms in Arabidopsis (Chao and Chao, 2022). In soil, protons released by $H^+$-adenosine triphosphatase increase iron solubility by acidifying the rhizosphere (Takeshige et al, 1988). Plants release phytochemicals of the coumarin family into the rhizosphere to chelate and mobilize $Fe^{3+}$ and reduce it to $Fe^{2+}$. This process helps in the absorption of iron by the plants.

Under conditions of iron deficiency, the basic-helix-loop-helix (bHLH) family FER-like iron deficiency-induced transcription factor (FIT) plays a crucial role in regulating the expression of iron uptake genes in the roots of Arabidopsis (Schwarz and Bauer, 2020). Activation of transcription of FRO2 and IRT1 genes occurs through simultaneous activation of a heterodimer composed of FIT and the Ib subgroup members of the bHLH transcription factor family (bHLH38, bHLH39, bHLH100, and bHLH101), which is triggered by signals indicating iron deficiency (Cai et al, 2021). The activation of the iron uptake gene improves the plant's ability to withstand iron deficiency. This activation is facilitated by the subgroup IVc bHLH transcription factors, namely bHLH34, bHLH121, ILR3 (bHLH105), and bHLH115, which activate the subgroup Ib genes to promote iron uptake (Lei et al, 2020). In circumstances characterized by limited iron availability, the removal of each gene belonging to the IVc subgroup results in the cessation of subgroup Ib gene activation and exacerbates the manifestation of iron deficiency symptoms (Lei et al, 2020). The regulation of gene activity achieved by the IVc subgroup may occur at the protein level.

Based on prior research, it has been established that the iron-binding E3 ligase BRUTUS (BTS) plays a crucial role in facilitating the degradation of IVc subgroup bHLH proteins (Choi et al, 2022). In the context of the *bts* mutant, the abundance of the IVc transcription factor protein was notably enhanced, leading to an upregulation in the protein abundance of the IVc subgroup. Consequently, this upregulation stimulated the expression of subgroup Ib genes, thereby activating the iron uptake gene expression (Hindt et al, 2017). Therefore, *bts* mutants are more

[1]Laboratory of Photosynthesis and Environment, CAS Center for Excellence in Molecular Plant Sciences, Shanghai Institute of Plant Physiology and Ecology, Chinese Academy of Sciences, Shanghai 200032, China. [2]University of Chinese Academy of Sciences, Beijing 100039, China. ✉E-mail: ppxu@cemps.ac.cn; hqzheng@cemps.ac.cn; wmcai@cemps.ac.cn

tolerant of iron deficiency. Biochemical experiments have confirmed that BTS interacts physically with either ILR3 or bHLH115 (Gao et al, 2020). The iron uptake genes expression was mitigated by introducing an allele from either the *ilr3* or *bhlh115* mutants into the *bts* mutant background. Compared to the *bts* single mutant, both the *btsbhlh115* and *btsilr3* double mutants exhibit decreased iron deficiency tolerance (Lei et al, 2020; Kurt et al, 2019). Iron toxicity was inhibited in the double mutant *ilr3bts*, indicating that *ILR3* gene repression prevented iron absorption.

The Arabidopsis DOF transcription factor family comprises 37 members, each of which plays a distinctive role in the growth and development. These members achieve their roles by binding to conserved DNA *cis*-elements, specifically CTTT(T/A) or (A/T) AAAG (Shimofurutani et al, 1998, Zou et al, 2023). The DOF protein harbors a conserved DNA binding-with-one-finger (DOF) domain, which encompasses a zinc-finger motif (Yanagisawa, 1997). OBP1 was initially identified based on its interaction with the OCS element-binding protein OBF4 (Chen et al, 1996). The other three subfamily proteins, namely OBP2, OBP3 and OBP4, were isolated by homolog analysis. No significant differences in DNA-binding properties were found between the different OBP proteins, although different OBP proteins have different functions in specific tissues in plants, even though they share similar protein–protein interaction properties. Plant cell cycle regulation is regulated by OBP1 (Skirycz et al, 2008), whereas the metabolism of indole glucosinolates is regulated by OBP2 (Skirycz et al, 2006). Previous studies reported that the *OBP3* expression level is regulated by salicylic acid, and that *OBP3*-overexpressed plants exhibit severe dwarfing, chlorosis and necrosis (Kang and Singh, 2000; Kang et al, 2003). However, it is not clear whether and, if so, how OBP3 regulates iron homeostasis in Arabidopsis and hence regulates plant development.

In the present investigation, the Arabidopsis transcription factor yeast one-hybrid (Y1H) library (Sivitz et al, 2012) was screened, leading to the identification of the DOF transcription factor OBP3 as the upstream regulator of the Ib subgroup *bHLH100* gene. This finding suggests the involvement of OBP3 in the iron-deficiency signaling pathway. The utilization of ChIP-qPCR and Y1H analysis demonstrated a direct correlation between OBP3 and the Ib subgroup bHLH genes promoters. Previous studies have indicated that the DOF protein, characterized by a single zinc-finger structure, commonly collaborates with other transcription factors to augment the expression specificity of targeted genes in vivo (Gao et al, 2020; Yamamoto et al, 2006; Gao et al, 2022a). The current study shows, as a consequence of iron deficiency, that OBP3 accumulates and interacts with the subgroup IVc bHLH ILR3. Then, the resulting heterodimer binds to the Ib bHLH subgroup gene promoters and promotes their expression. This study elucidates the mechanism through which ILR3 facilitates the activation of specific target genes by OBP3, thereby regulating iron homeostasis and plant growth. At the posttranscriptional level, the E3 ligase BRUTUS (BTS) plays a role in promoting the degradation of the OBP3 protein via the 26S proteasome machinery, thereby preventing excessive iron uptake and maintaining iron homeostasis in plants. Consequently, this research uncovers a novel mechanism by which OBP3 finely tunes plant iron homeostasis.

# Results

## Identification of the DOF family member OBP3 as a *bHLH100* promoter-interacting protein in Arabidopsis

The full-length promoter of the Ib subgroup bHLH gene *bHLH100* extends 1541 bp upstream of the starting codon (pro1541), signals the availability of Fe and is significantly induced by Fe starvation (Fig. 1A). For identification of the upstream regulatory factor, we first identified the minimum fragment of the *bHLH100* promoter required for Fe availability response using a 5'-deletion series of the *bHLH100* promoter with the GUS reporter gene (Fig. 1A). Arabidopsis transformants carrying the pro1541::GUS (pro1541G), pro1117G, pro690G and pro320G constructs generated high GUS signals under Fe-deficiency conditions, whereas pro320G did not respond to Fe starvation (Fig. 1B). This indicated that pro690 contained the minimum sequence required to respond to Fe availability-responsive expression.

In the Y1H screening experiment, the pro690 fragment was used as the bait to identify the upstream TF regulating *bHLH100* expression (Fig. 1A–C). A bait reporter yeast strain containing the pro690 construct was transformed using a prey library composed of 1589 TFs in *Arabidopsis* (Ji et al, 2018; Ou et al, 2011). We determined sequence of the prey plasmid from nine positive colonies and found that seven of them had previously been reported to be iron deficiency-responsive *bHLH* family members AT5G54680 (*bHLH105/ILR3*), AT1G51070 (*bHLH115*), and AT3G47640 (*PYE*) (Fig. 1D; Appendix Table S1). Interestingly, the remaining clone belonged to the DOF family transcription factor OBP3 (Ward et al, 2005). Arabidopsis genome has 37 DOF family members, and the OBP3 protein is part of a subgroup with four members within a smaller clade (Yanagisawa, 2002) (Appendix Fig. S1). As their name indicates, the DOF TF OBF BINDING PROTEINs (OBPs) were first identified as the bZIP TF OCS BINDING FACTOR 4 partner protein (Zhang et al, 1995). The Y1H experiment confirmed the interaction of the *bHLH100* promoter with *OBP3*, but not with other OBP subgroup proteins (Fig. 1D). To further test the interactions between the *bHLH100* promoter and either OBP3 or its related DOF TFs, we conducted the firefly luciferase (fLUC) assay using the DOF members of the OBP subfamily. Our transient assays in 2-week-old Arabidopsis seedlings showed that the OBP3 could increase fLUC activity under the control of the pro690 fragment of the *bHLH100* promoter (Fig. 1E); no other member of the OBP DOF family (OBP1, OBP2, OBP4) or DAG1, CDF1 achieved this effect. This result indicated that the OBP3 protein was directly associated with the *bHLH100* promoter. In the subsequent research, we focused on the characterization of the *OBP3* gene in response to iron deficiency.

## Expression level of *OBP3* depends on iron availability

The *OBP3* expression has been detected in various tissues of plants, and the abundance of *OBP3* mRNA in leaves and roots in the present study was relatively high (Ward et al, 2005). According to the quantitative real-time PCR (qPCR) analysis (Fig. 1F), iron deficiency upregulated *OBP3* expression. OBP3 promoter was cloned and fused with the GUS reporter gene (proOBP3::GUS). The activity of the OBP3 promoter was evaluated in seedlings roots

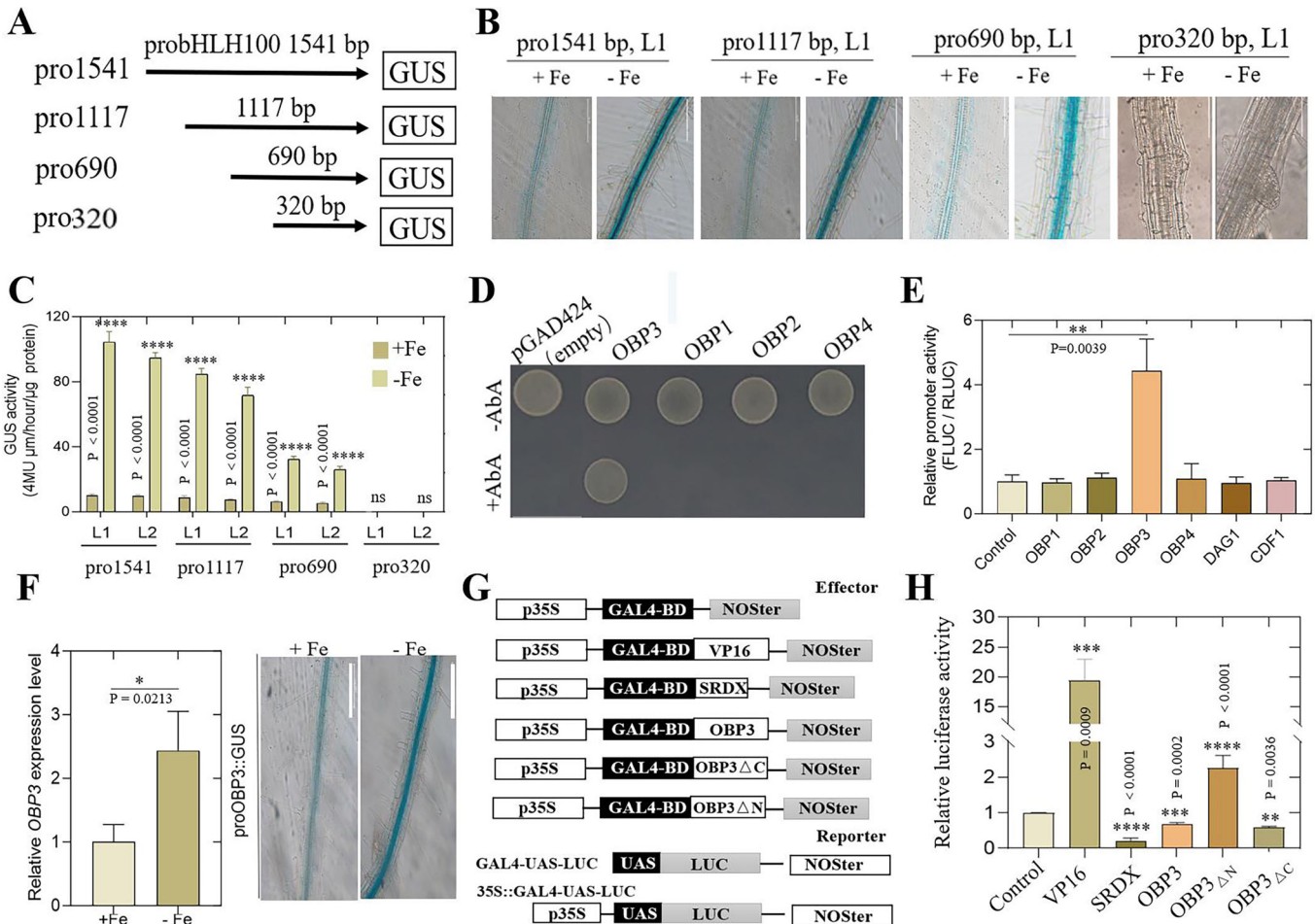

**Figure 1. Isolation and characterization of the DOF Protein OBP3 that can interact with the deleted *bHLH100* promoter.**

(A) This figure shows a schematic diagram of the full-length and truncated proOBP3::GUS fusion constructs. The 1541 bp OBP3 promoter fused with GUS is represented by pro1541. The deletion construct is named according to the length of promoter region. (B) and (C) show GUS staining and relative GUS activity in transgenic seedlings (L1 and L2) containing full-length or truncated *bHLH100* promoter::GUS constructs. The precise methodology for quantifying GUS activity detection is elaborated upon in the Methods section. The seedlings, which were 5 days old, were transferred for 3 days on half-strength MS plates with 50 μM $Fe^{3+}$ (+ Fe) or 0.5 μM $Fe^{3+}$ (− Fe). A representative image is presented, and the error bar represents SD ($n = 3$). Bar indicates 100 μm. Asterisks indicate significant differences (****$P < 0.0001$; Student's $t$-test). Exact $P$ values are provided in the figure. (D) Y1H analysis was conducted to investigate the interaction between promoters of DOF TF members of the OBP subfamily, using the bHLH100 pro690 promoter as bait. The OBP3 cDNA was cloned into pGAD424 as prey, allowing the transformed yeast strains to grow on the selective medium. (E) Interaction between DOF members of OBP subfamily and the bHLH100 promoter was assessed in a transient assay using the pro690::fLUC construct, which carries the bHLH100 pro690 promoter fused with the luciferase gene, as the reporter gene. The vector control or selected DOF family TFs as effectors. Promoter activity was measured by the ratio of fLUC: RLUC activity. The internal control used was the CaMV 35S promoter vector fused with the RLUC reporter gene. The error bar represents SD ($n = 3$). Asterisk denotes significant differences (**$p < 0.01$; Student's $t$-test). Exact $P$ values are provided in the figure. (F) OBP3 expression in WT plants grown on Fe sufficient medium (+ Fe) for 5 days was analyzed using qPCR. The plants were then transferred to Fe deficient or +Fe medium for 48 h. After 48 h of iron deficiency treatment, proOBP3::GUS induction was observed in roots. Error bar represents SD ($n = 3$). The bar is 100 μm. Asterisks indicate significant differences (*$P < 0.05$; Student's $t$-test). The reporter and effector construction are shown in (G, H). The reporter gene used in this study is GAL4-*LUC*. The effector constructs contained the GAL4 DNA-binding domain fused with either the wild type or truncated OBP proteins (GAL4-OBP3 or GAL4-OBP3$_{\Delta N}$ or GAL4-OBP3$_{\Delta C}$). The controls used were GAL4 and GAL4-VP16. Transcription activation of GAL4-OBP fusion protein was detected using Arabidopsis protoplasts. The *LUC* reporter gene and multiple effector genes were co-transfected into Arabidopsis protoplasts. The control is set to 1. Error bar represents SD ($n = 3$). Asterisks indicate significant differences (**$P < 0.01$, ***$P < 0.001$, ****$P < 0.0001$; Student's $t$-test). Exact $P$ values are provided in the figure. Source data are available online for this figure.

grown under iron sufficiency and iron deficiency conditions. Figure 1F shows that GUS activity in Arabidopsis seedlings roots grown under iron deficiency was obviously higher than that under iron sufficiency.

As the OBP3 protein interacted with the bHLH100 promoter, we investigated whether the upregulated expression of the Ib subgroup bHLH genes, *bHLH38*, *bHLH39*, *bHLH100*, and *bHLH101*, in response to iron deficiency, was affected by OBP3.

A T-DNA insertion line for OBP3 (*obp3-2*, CS853401) was ordered from the NASC. Using 7-d-old Arabidopsis seedlings, exposure to Fe-deficiency treatment for 24 h caused a significant upregulation of the expression of Ib subgroup bHLH genes by 80- to 160-fold, whereas in the *obp3-2* mutant background, the expression of these genes was only upregulated by 40- to 60-fold (Appendix Fig. S2). Fe deficiency induced upregulation of these bHLH1b family genes in the roots at least partially depends on the function of the *OBP3*

gene. So, it was clear that Fe availability affected *OBP3* expression and upregulated Ib subgroup bHLH genes expression in response to iron deficiency was through OBP3 cascade.

We used a transient expression assay in protoplasts to test how OBP3 performs its transcription regulatory activity through the fusion with GAL4 transcriptional activator. Our experiments involved transforming reporter gene and effector gene constructs into protoplasts prepared from rosette leaves. The transcriptional activation induced by different fused proteins was examined. Figure 1G shows the wild type and terminal deletion of OBP3 fused with the GAL4 transcriptional activator. The transcriptional activation was monitored by reporting the UAS-GAL4 cassette upstream of the promoter of the *luciferase* gene. It was observed that the trans-activation activity of the GAL4-OBP3 fusion proteins was lower than that of the single GAL4-binding domains used as the control (Fig. 1H). Therefore, we did not see evidence of trans-activation constructs with the full-length OBP3 protein. Compared with the GAL4-VP16 fusion protein, the GAL4-OBP3 fusion protein with the N-terminal deletion can trans-activate the reporter gene expression. According to our experiment, sequences containing residues 181–368 without the N-terminal region of OBP3 caused transcriptional activation (Fig. 1H). However, sequences containing residues 1–180 inhibited the luciferase activity (Fig. 1H). Research has demonstrated that the OBP3 N-terminal region, which includes the DOF domains, can inhibit protein's trans-activation activity. In addition, Arabidopsis CDF4, maize DOF2, and barley PBF have the ability to function as either transcriptional activators or repressors, depending on their interaction partners (Xu et al, 2020; Yanagisawa 2000; Yamamoto et al, 2006).

## The overexpression of *OBP3*, whether constitutive or inducible, results in iron overload in transgenic plants

Through a literature search, we found that previous work has shown that overexpression of the *OBP3* gene can lead to leaf necrosis and severe growth retardation (Kang and Singh, 2000; Kang et al, 2003). However, the mechanism behind this is not clear. Given that our work suggests that the *OBP3* gene responds to iron deficiency, it is strongly suggested that the growth arrest phenotype caused by *OBP3* overexpression may result from an imbalance in iron homeostasis in the plant. We also generated constitutively *OBP3*-overexpressed plant. The findings are consistent with previous reports (Fig. 2A), with chlorosis and necrotic lesions being found in the rosette leaves of 35S::*OBP3* transgenic plants (Fig. 2B, red arrow) (Kang and Singh, 2000; Kang et al, 2003). In addition, the leaves of 35S::*OBP3* plants also contained a lower chlorophyll concentration and a significant higher concentration of ferric ions (Fig. 2B,D). Furthermore, the root length was found to be significantly reduced in 35S::*OBP3* plant seedlings (Appendix Fig. S3A). As a result of these toxic symptoms, we used the Perls' Prussian blue (PPB) staining method to detect $Fe^{3+}$ in 35S::*OBP3* plants and compared the results with those in wild-type plants. The $Fe^{3+}$ staining was very weak in wild-type rosette leaves, whereas 35S::*OBP3* leaves exhibited high levels of $Fe^{3+}$, as illustrated in Fig. 2C. The trichomes of 35S::*OBP3* rosette leaves also showed high levels of stainable $Fe^{3+}$ (Fig. 2C). In 35S::*OBP3* plants, the *FERRETIN1* (*FER1*) and *FERRETIN2* (*FER2*) genes demonstrated significantly higher transcript abundance levels than in the control plants (Appendix Fig. S3B). Iron overload resulted in an increase in

expression of ferritin genes, which protected the cells from oxidative damage caused by excessive iron concentrations. This indicated that *OBP3* overexpression resulted in the excessive uptake of iron, which may be retained in 35S::*OBP3* plants.

To further elucidate the direct impact of *OBP3* on iron accumulation and at the same time, avoid the severe defects caused by *OBP3* overexpression, we used an estrogen-inducible promoter to control *OBP3* expression and to observe the phenotype of pER8::*OBP3* transgenic plants grown in soil in response to estrogen treatment (Appendix Fig. 3C) (Xu et al, 2020). No differences from control plants were observed in pER8::*OBP3* plants without estrogen treatment during the 5-week growth period. However, after 1-week treatment with 30 μM β-estrogen, chlorosis and necrosis occurred in leaves and some siliques near the base of the peduncles of pER8::*OBP3* transgenic plants (Fig. 2E). We used Perls' staining to detect whether the iron concentration was significantly different between them. We found that the pER8::*OBP3* transgenic plants showed significant signs of $Fe^{3+}$ staining, including flowers and siliques, compared with the corresponding organs of without estrogen treatment plants (Fig. 2F).

## The *obp3* knockout mutants displayed altered Fe homeostasis

The bHLH family transcription factors are important in maintaining Fe homeostasis in plants, according to previous studies (Cui et al, 2018). However, prior to this study, no DOF family zinc-finger proteins had been reported to be involved in iron homeostasis. To confirm the role of the OBP3 protein, two homozygous OBP3 T-DNA insertion mutants were obtained, one of which had T-DNA inserted into the 3′ UTR (3′ untranslated region) (*obp3-1*) and the other with T-DNA inserted into the exon (*obp3-2*) (Fig. 3A), with approximately 82% knockdown and complete knockout of expression, respectively (Fig. 3B). In the sufficient iron medium (50 μM Fe), no significant difference growth phenotype was observed between *obp3* mutant and control plants. However, when the plants were grown on a medium deficient in iron (0.5 μM Fe), the *obp3* mutant exhibited significant inhibition of root length, the development of chlorotic leaves, and a decrease in the content of ferric ions and fresh weight compared to the control plants (Fig. 3C–F). In addition, the expression of the *FRO2* (FERRIC REDUCTION OXIDASE 2) and *IRT1* (IRON-REGULATED TRANSPORTER 1) genes is downregulated in the *obp3* mutant (Appendix Fig. S4A). This may contribute to a reduction in the activity of ferric-chelate reductase in the *obp3* mutant root (Appendix Fig. S4B), resulting in iron deficiency phenomenon in the plant.

Next, we examined the phenotypes of soil-grown *obp3* mutants (Fig. 3G). Iron concentration in *obp3* mutant young leaves was only 9% lower than that of Col-0 when grown in soil with normal pH (about pH 5–6). However, an alkaline environment (about pH 7–8) made it easier to form insoluble ferric hydroxide ($Fe(OH)_3$), which cannot be absorbed by the plants. The *obp3* mutant showed overall growth stagnation after cultivation under high pH conditions, with several bleached and smaller true leaves (Fig. 3G). The iron concentration and chlorophyll content of the *obp3* mutant grown in alkaline soil were lower than those of the control (Fig. 3H,I). These findings indicate that *OBP3* plays a critical role in efficient iron absorption during iron deficiency conditions.

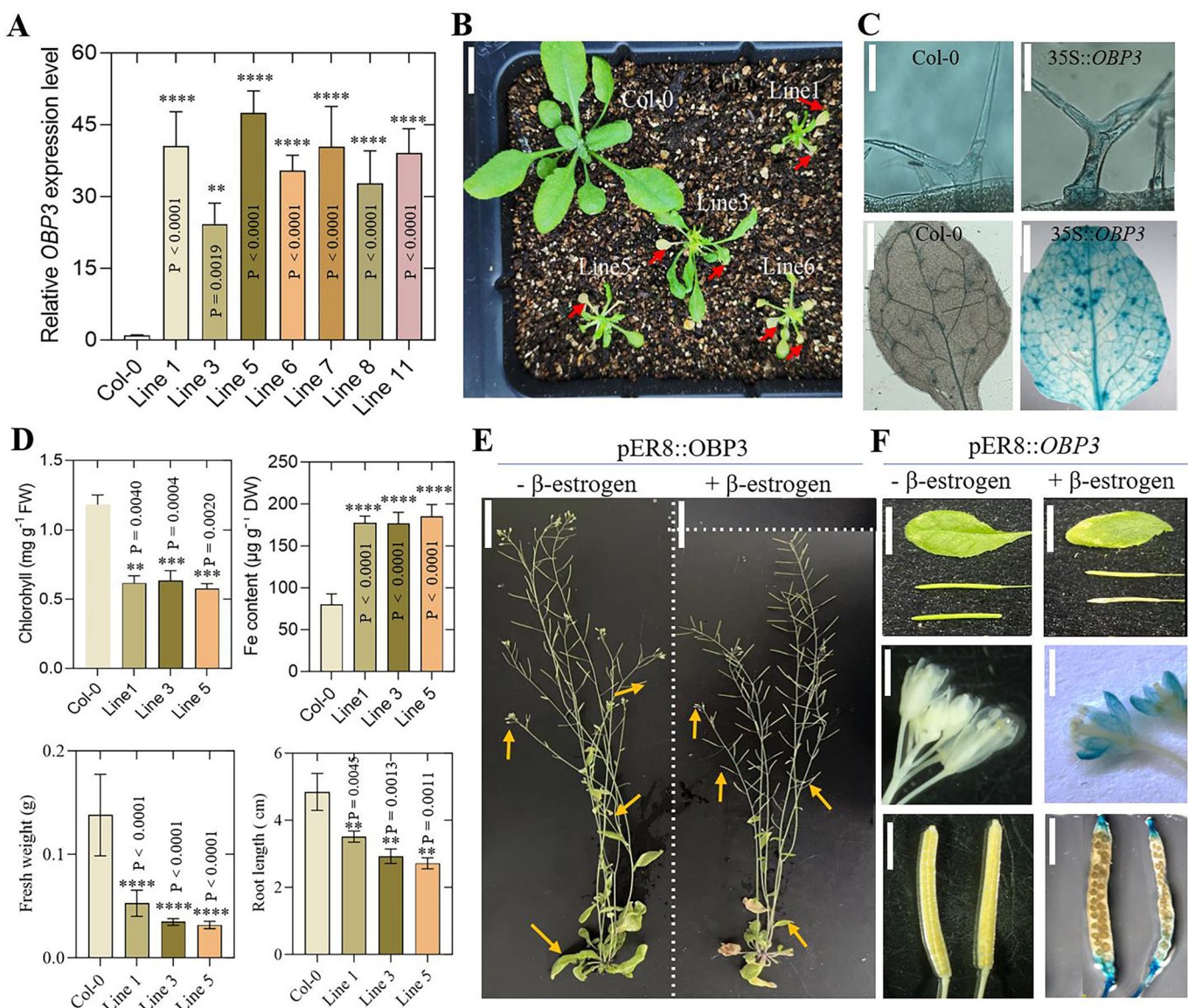

**Figure 2.   The constitutive and inducible overexpression of *OBP3* leads to excessive accumulation of iron.**

(A) This figure shows *OBP3* expression level in Col-0 and 35S::*OBP3* transgenic plants. The error bar represents the standard deviation. n = 3. Asterisk denotes significant differences (***$p < 0.001$, ****$p < 0.0001$; Student's *t*-test). Exact *P* values are provided in the figure. (B) This image displays the wild-type and 35S::*OBP3* transgenic plants rosette leaves. The scale bar is 1 cm. (C) This figure shows the Perls' Fe staining signals in both wild-type and 35S::*OBP3* rosette leaves and trichomes. We examined three independent 35::*OBP3* transgenic lines, each with three rosette leaves with similar results. A typical picture of the staining results is shown. The scale bar is 100 μm for trichomes and 0.2 cm for rosette leaves. (D) This study aimed to detect relative chlorophyll content, iron concentration, and fresh weight in wild-type and 35S::*OBP3* transgenic plants rosette leaves. The error bar represents SD, n = 3. In addition, root length was measured in 7-day-old wild-type and 35S::*OBP3* transgenic plants. The error bar represents SD, n = 3. Asterisk denotes significant differences (**$p < 0.01$, ***$p < 0.001$, ****$p < 0.0001$; Student's *t*-test). Exact *P* values are provided in the figure. (E) The phenotype of 5-week-old pER8::OBP3 plants before and after exposure to 30 μM estrogen for 1 week. Bar = 2 cm. (F) the observation of aborted silique and yellowed leaves, and Perls' Fe staining signals in wild-type and pER8::*OBP3* (marked with yellow arrows in (E)) flowers and siliques. Bar = 0.4 cm. Source data are available online for this figure.

## OBP3 directly activates transcription of Ib subgroup genes *bHLH38/39/100/101*

Previous studies have shown that the bHLH 1b family transcription factors *bHLH38* and *bHLH39* (previously named *ORG2* and *ORG3*) are putative downstream targets of *OBP3* (Kang and Singh, 2000; Kang et al, 2003). Considering the absence of experimental

evidence regarding direct regulation in vitro and in vivo in early studies conducted two decades ago. Our study also indicates that *OBP3* is an upstream regulator of *bHLH100*. These findings strongly suggest that OBP3 plays a crucial role in regulating iron homeostasis in plants by directly regulating bHLH 1b family genes. The Ib subgroup of *bHLH* genes promoter contained numerous clusters of DOF transcription factor regulatory elements (Fig. 4A).

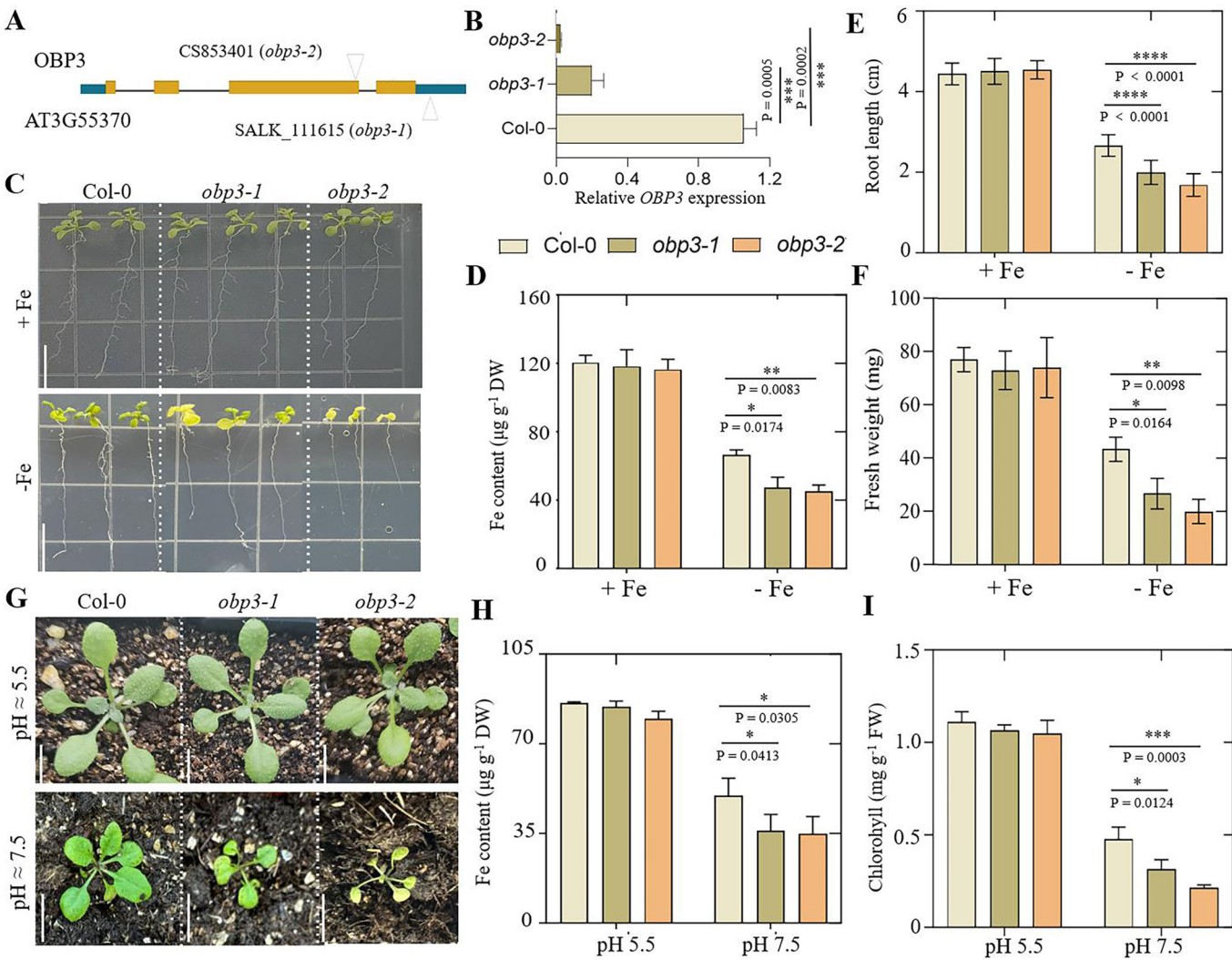

**Figure 3. The *obp3* mutants exhibit reduced tolerance to Fe-deficient conditions.**

(A) The T-DNA insertion sites of *obp3-1* (SALK_111615) and *obp3-2* (CS853401) mutants are shown. (B) *OBP3* transcripts in seedlings of Col-0, *obp3-1*, and *obp3-2* mutants were detected by using qPCR method. Error bar represents SD. *Significant differences ($p < 0.05$; Student's t-test) were observed. (C) The phenotypes of Col-0 and the *obp3* mutants were observed after being grown for 11 days on Fe-sufficient (+ Fe, 50 µM Fe) and Fe-deficient (− Fe, 0.5 µM Fe) media. The bar measures 1.5 cm. (D) The iron concentrations, (E) root length, and (F) fresh weight in seedlings grown on Fe-sufficient or Fe-deficient media in (C) were measured. For root length ($n = 10$) and for iron concentration and fresh weight ($n = 3$). The data represent the means ± SD of three independent experiments. (G) The phenotypes of 3-week-old wild type and *obp3* mutants grown on either normal soil (pH 5 to 6) or alkaline soil (pH 7 to 8). Bar represents 1 cm. (H) Analysis of iron concentrations and (I) chlorophyll content in rosette leaves of wild type and *obp3* mutants grown on normal or alkaline soil is presented in (E). $n = 3$. The data represents the mean ± SD. DW, dry weight. For (D–F, H, I), data were analyzed by two-tailed Student's t-test. Asterisk denotes significant differences (*$p < 0.05$, **$p < 0.01$, ***$p < 0.001$, ****$p < 0.0001$; Student's t-test) (D–F, H, I). Exact *P* values are provided in the figure (D–F, H, I). Source data are available online for this figure.

We conducted comparative qPCR analysis of major genes involved in the iron deficiency pathway. Under iron deficiency conditions, the Ib subgroup bHLH genes exhibited significantly lower expression levels in *obp3* mutant than in control. This suggests that *OBP3* is a key upstream activator of these transcription factors (Appendix Fig. S5). Consistent with these findings, the constitutive overexpression of *OBP3* significantly increased the Ib subgroup *bHLH* genes expression, indicating that *OBP3* regulates iron homeostasis upstream of these genes (Fig. 4B). However, the IVc subgroup bHLH genes expression level, such as *bHLH34 bHLH115*, and *PYE*, were similar in *obp3* mutant and Col-0 (Appendix Fig. 5).

In the following step, we conducted chromatin immunoprecipitation (ChIP)-qPCR assay to identify whether OBP3 can bind to target genes promoter. The experiments were performed using the 35S::*OBP3*-GFP transgenic construct. The ChIP-qPCR results demonstrate that OBP3 can bind to the *bHLH38*, *bHLH39*, *bHLH100*, and *bHLH101* genes promoters in vivo (Fig. 4C).

Simultaneously, constructs were created to generate the dexamethasone (DEX) inducible OBP3 transgenic plants. In vitro treatment with cycloheximide (CHX), which blocks eukaryotic translation and protein synthesis, and DEX showed a significant upregulation of *OBP3* expression. The expression of its candidate target genes was significantly upregulated within 1–3 h of treatment (Fig. 4D). This suggest that the

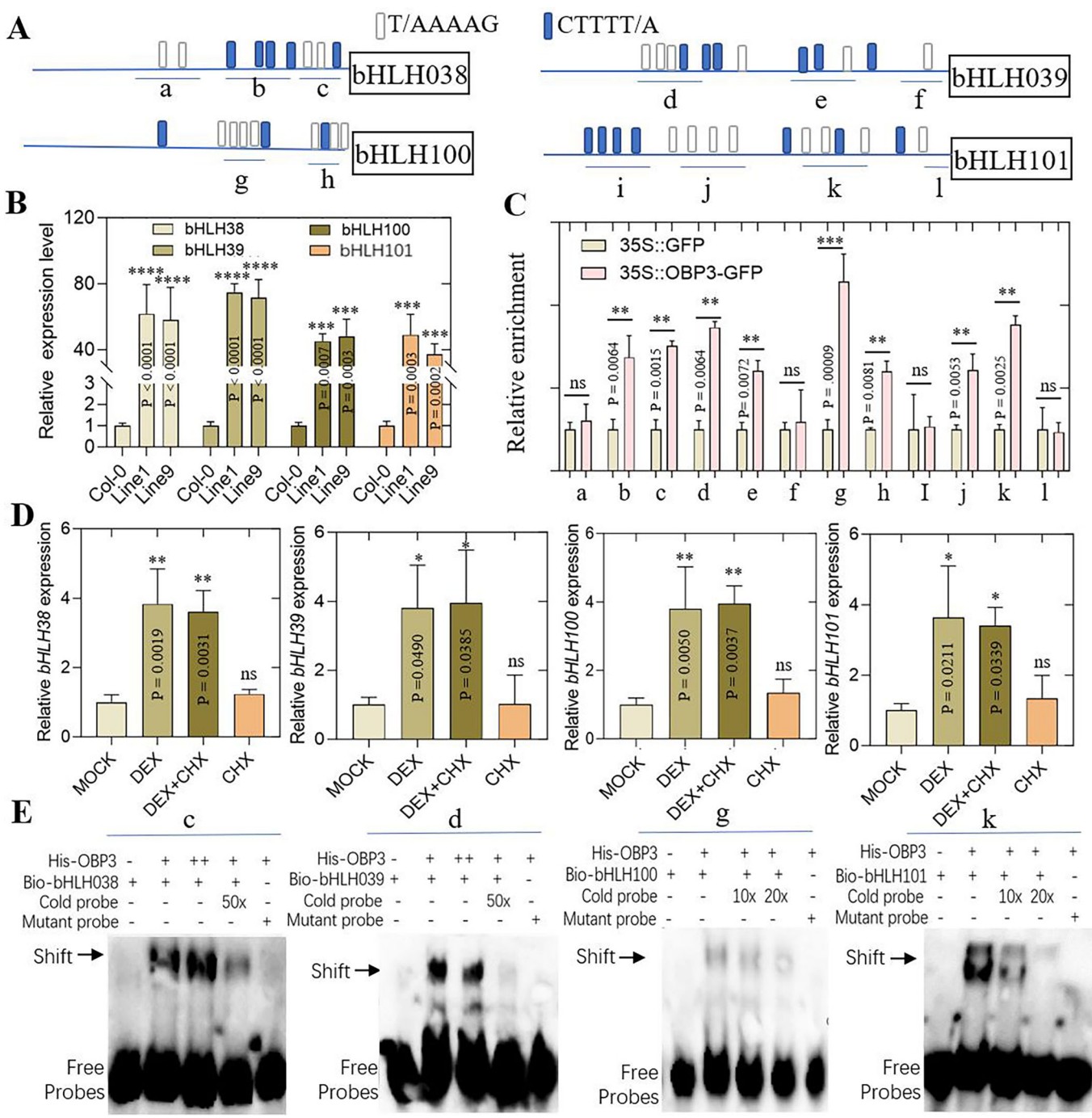

downstream target genes expression activated by *OBP3* does not depend on the synthesis of new proteins. In addition, we conducted EMSA experiments with the OBP3 protein expressed in vitro. The results demonstrate that the OBP3 protein can bind to the promoter regions of the selected target genes in vitro (Fig. 4E; Appendix Fig. S6). This indicates that *OBP3* directly regulate the expression of the *bHLH 1b* genes.

## OBP3 interacts with ILR3 in vitro and in vivo

Y2H screening assays were conducted to identify potential cofactors that interacted with OBP3 in controlling iron

homeostasis (Zhang et al, 1995). Out of all the transformants, many putative interactors were obtained. ILR3, a bHLH family transcription factor, was among the potential OBP3-interacting proteins (Samira et al, 2018) (Appendix Table S2). To confirm the Y2H screening results, we verified the direct interactions between them. OBP3 was used as a bait (binding domain fusion with GAL4) and ILR3 as a prey (activating domain fusion with GAL4) (Fig. 5A). In addition, we tested the interaction between OBP3 and other bHLH proteins, including bHLH121, bHLH115, PYE, and bHLH39 (Fig. 5A). The Y2H assay results showed that only ILR3 could interact with OBP3.

**Figure 4. OBP3 directly binds to the promoters of *bHLH38/39/100/101*.**

(A) Schematic diagram showing the *bHLH38/39/100/101* gene promoter regions and the location of sense ([T/A]AAAG, white rectangle) and reverse (CTTT[A/T], blue rectangle) DOF binding motifs. (B) qPCR analysis of 1b subgroup *bHLH* genes in Control and 35S::*OBP3* transgenic plants. Value indicates mean ± SD, $n = 3$. Data were analyzed by two-tailed Student's $t$-test. Asterisk denotes significant differences (***$p < 0.001$, ****$p < 0.0001$; Student's $t$-test). Exact $P$ values are provided. (C) ChIP assays showing OBP3 binding to the 1b subgroup *bHLH* genes promoters in vivo. Three-week-old 35S::*OBP3*-GFP or 35S::GFP seedlings were harvested for ChIP-qPCR analysis, using anti-GFP antibody, and the precipitated DNA was analyzed by qPCR assays. a–l represents the different promoter segments selected in (A). In the various promoter segments, the 35S::GFP control was designated as the reference value of 1. Data are means ± SD, $n = 3$. Data were analyzed by two-tailed Student's $t$-test. Asterisk denotes significant differences (****$p < 0.0001$; Student's $t$-test). Exact $P$ values are provided. (D) *bHLH38*, *bHLH39*, *bHLH100*, and *bHLH100* relative expression levels in 8-d-old *OBP3*-GR transgenic plants treated with 25 μM DEX, 80 μM CHX, 25 μM DEX plus 80 μM CHX or mock (control). Values are given as mean ± SD, $n = 3$. Data were analyzed by two-tailed Student's $t$-test. Asterisk denotes significant differences (****$p < 0.0001$; Student's $t$-test). Exact $P$ values are provided. (E) EMSA showed that OBP3 binds to *bHLH38* (fragment "c"), *bHLH39* (fragment "d"), *bHLH100* (fragment "g"), and *bHLH101* (fragment "k") promoters. Biotin-labeled DNA fragments were incubated with His-OBP3, and competition assays for the labeled promoter sequences were performed by adding an excess of unlabeled cold probes. Two biological replicates were performed, giving similar results. Source data are available online for this figure.

We conducted various methods to confirm the direct interaction between OBP3 and ILR3 in plants. A Bimolecular Fluorescence Complementation (BiFC) assay was conducted. The strong fluorescent signals only in the nuclei of plants where OBP3-cYFP was co-expressed with ILR3-nYFP in the tobacco leaves epidermis cell (Fig. 5B). This suggests that interaction between OBP3 and ILR3 occurs in the nucleus. The interaction between OBP3 and ILR3 was confirmed by co-immunoprecipitation assay using the co-expression of 35S::OBP3-myc and 35S::ILR3-GFP in transgenic tobacco plants (Fig. 5C). These data indicate that OBP3 forms a heterodimer with ILR3.

## Binding of ILR3[bHLH] to *bHLH100* enhances OBP3[DOF] binding and ILR3 enhances the transcriptional activity of *OBP3*

Like the conventional DOF family zinc-finger protein (ZF), OBP3 has a highly disordered structure beyond the DOF DNA-binding domain. It is predicted that the OBP3[DOF] domain contains one helix and two β-sheets (Fig. 6A). In the conserved motif CX2CX21CX2C, four cysteine residues may bind metal ions (Zn) (Fig. 6A). We generated the OBP3[DOF] DNA-binding domain (OBP3[DOF]) fused with the N-terminal MBP (maltose-binding protein), with the MBP-OBP3[DOF] protein being purified from *E. coli*. Then, we mutated the OBP3[DOF] protein sequence to verify the model. It was predicted that the binding of metal ions to the four cysteine residues would result in a very stable interaction between the OBP3[DOF] domain and the DNA sequence (Gao et al, 2022a). In line with this, mutation of each of the four cysteine residues (C124, C132, C146, and C149) to Alanine (OBP3[DOF] (mu1)) eliminated the interaction between OBP3[DOF] and DNA (Fig. 6A). The α-helices of OBP3[DOF] can be embedded into the groove of DNA and facilitated DNA binding. Mutations in Y148 (mu2: T152A) or W150 (mu3: R153A) downregulated its DNA-binding activity. Mutations in R151 (mu4: R151A), Y152 (mu5: Y152A), or W153 (mu6: W153A) in the helix eliminated its DNA-binding activity (Fig. 6A), which is similar with the conserved residues in DOF AOBP reported previously (Shimofurutani et al, 1998), that are involved in DNA recognition. As with mu7 (C146A and C149A), mutations in the structural domain of the OBP3[DOF] eliminated its DNA-binding activity (Fig. 6A). This study confirmed that these residues were involved in DNA binding. EMSA analysis was then performed to identify the DNA sequence that interacted with OBP3[DOF] (Fig. 6B). EMSA results showed that 5-bp [T/A]AAAG motif displayed the most effective binding activity. Hence, AAAAG was more likely to be recognized by OBP3 than was AAAG, and the position of 3'-G determined the binding affinity of OBP3[DOF] with DNA in vitro.

Zinc-finger (ZF) proteins usually contain several ZF motifs arranged in tandem, which can increase the protein's interaction with DNA. However, the DOF proteins, including OBP3, contain only single ZF motifs. We therefore speculated that the interaction with ILR3 may improve the binding specificity of OBP3 to the "[A/T]AAAG or CTTT[T/A]" motifs. We further wanted to test whether ILR3[bHLH] enhanced the binding of OBP3[DOF] in vitro. In Fig. 6C b.1, a supershift was observed, indicating that they could bind to the same *bHLH100* fragment, and binding was significantly reduced by mutations in "CACGTG" G-boxes (Fig. 6C b.2). This suggested that the binding of ILR3[bHLH] to the G-box was the determinant of supershift. In the absence of both G-box and DOF binding motifs as a result of mutations, no supershift was detected (Fig. 6C b.3). In this way, ILR3[bHLH] is required for OBP3[DOF] to bind DNA in order to supershift. Furthermore, ChIP-qPCR analysis indicated that OBP3 binding to DNA exhibited a strong association with the G-box (Fig. 6D). Mutating *ILR3* gene can significantly down-regulate the regulatory ability of OBP3 on the target gene *bHLH100* (Fig. 6D). Consistent with this observation, the *bHLH100* gene and other bHLH1b subgroup genes expression levels are lower in the *obp3ilr3* double mutant background compared with the single *obp3* mutant (Fig. 6D; Appendix Fig. 7).

OBP3 is a transcriptional repressor that activates downstream iron-deficiency signaling genes. We used a transient expression assay to identify whether the OBP3/ILR3 interactions affect the OBP3 transcriptional activity. To test our hypothesis, we designed an in vivo transcription assay. ILR3 was fused with the GAL4 DNA-binding domain under 35S promoter (Fig. 6E). The reporter gene construct comprises a minimal 35S promoter and the GAL4 DNA-binding site that drives the LUCIFERASE (*LUC*) reporter gene. We transiently expressed these constructs in Arabidopsis protoplasts and detected the *LUC* reporter gene activity. Relative to the control group, expression of OBP3 alone inhibited transcriptional activity, whereas simultaneous expression of ILR3 and OBP3 proteins enhanced the transcriptional activation activity. So, the transcriptional activation function of OBP3 may require the assistance of ILR3 protein (Fig. 6E). Accordingly, we propose a model map of OBP and ILR3 protein interactions that synergistically regulate iron homeostasis in plants (Fig. 6F).

## The *obp3ilr3* double mutant enhances susceptibility to iron deficiency, compared with the single mutant, and overexpression of *bHLH100* complements the mutant's iron-deficient phenotype

Because of the interaction between OBP3 and ILR3 proteins and their joint role in regulating iron homeostasis, to genetically

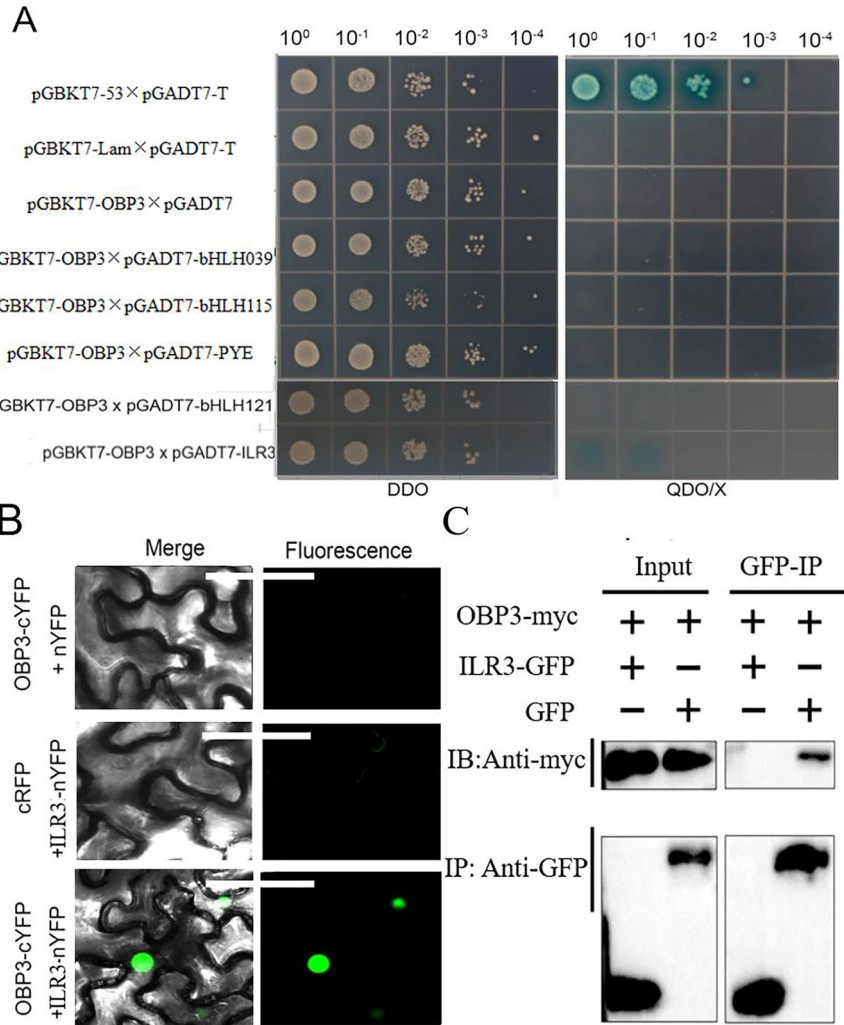

**Figure 5. OBP3 interacts with ILR3 in vivo and in vitro.**

(A) Y2H experiment was conducted to investigate the interaction between OBP3 and ILR3 (bHLH105), bHLH39, PYE, bHLH115, or bHLH121 proteins. The positive control was pGBKT7-53×pGBKT7-T, while the empty control was pGBKT7-LAM×pGBKT7-T. The bait (pGBKT7-53/-LAM/-OBP3) and prey (the indicated bHLH member cloned into pGADT7) pairs were transformed into yeast cells, and selected on SD-Ade/-Trp/-Leu/-His medium. The yeast is diluted over a concentration gradient and then spotted onto a plate. (B) BiFC analysis was used to investigate OBP3 and ILR3 interaction in *benthamiana* leaves epidermis cell. Represented fluorescence images and fluorescence images merged with light-view images were showed. Bar indicates 50 μm. (C) A co-immunoprecipitation experiment was conducted to investigate OBP3 and ILR3 interaction. *N. benthamiana* leaves co-expressing OBP3-Myc and ILR3-GFP or OBP3-Myc and the empty GFP vector were used to extracted the proteins. Immunoprecipitation was performed using c-Myc antibody-conjugated agarose beads. The precipitated proteins and the input samples were detected using anti-GFP or anti-Myc antibodies. Source data are available online for this figure.

confirm that the two proteins were involved in the same pathway, we obtained the *obp3ilr3* double mutant. By examining the growth of the double mutant in both iron-sufficient and iron-deficient media, we found that *obp3ilr3* mutant was more sensitive to low iron than was *obp3* mutant (Fig. 7A–C). Therefore, OBP3 and ILR3 acted synergistically to regulate iron homeostasis. Given our finding that OBP3 and ILR3 proteins target the transcription of the downstream *bHLH100* gene, we investigated whether increased *bHLH100* expression in the *obp3ilr3* double mutant inhibited its iron-deficiency-sensitive phenotype. We obtained the *obp3ilr3* & 35S::*bHLH100* plants. We found that overexpression of *bHLH100* partially recovered the inhibited root growth phenotype of *obp3ilr3*, with a reduced number of chlorotic leaves and a higher level of Fe

concentration in transgenic plants grown on either Fe-deficient medium or on alkaline soil (Fig. 7D–F). Together, these results suggested that increased expression of the downstream *bHLH* subgroup family gene *bHLH100* suppressed the iron-deficiency-sensitive phenotype of the *obp3ilr3* mutant.

## BRUTUS (BTS) targets OBP3 for degradation

Based on our previous Y2H screening analysis, the BRUTUS (BTS) protein, an important E3 ligase protein that involved in iron deficiency response, is also listed as a potential interacting protein for OBP3 (Appendix Table S2). It is hypothesized that OBP3 proteins may be downstream targets of BTS, as both BTS and OBP3

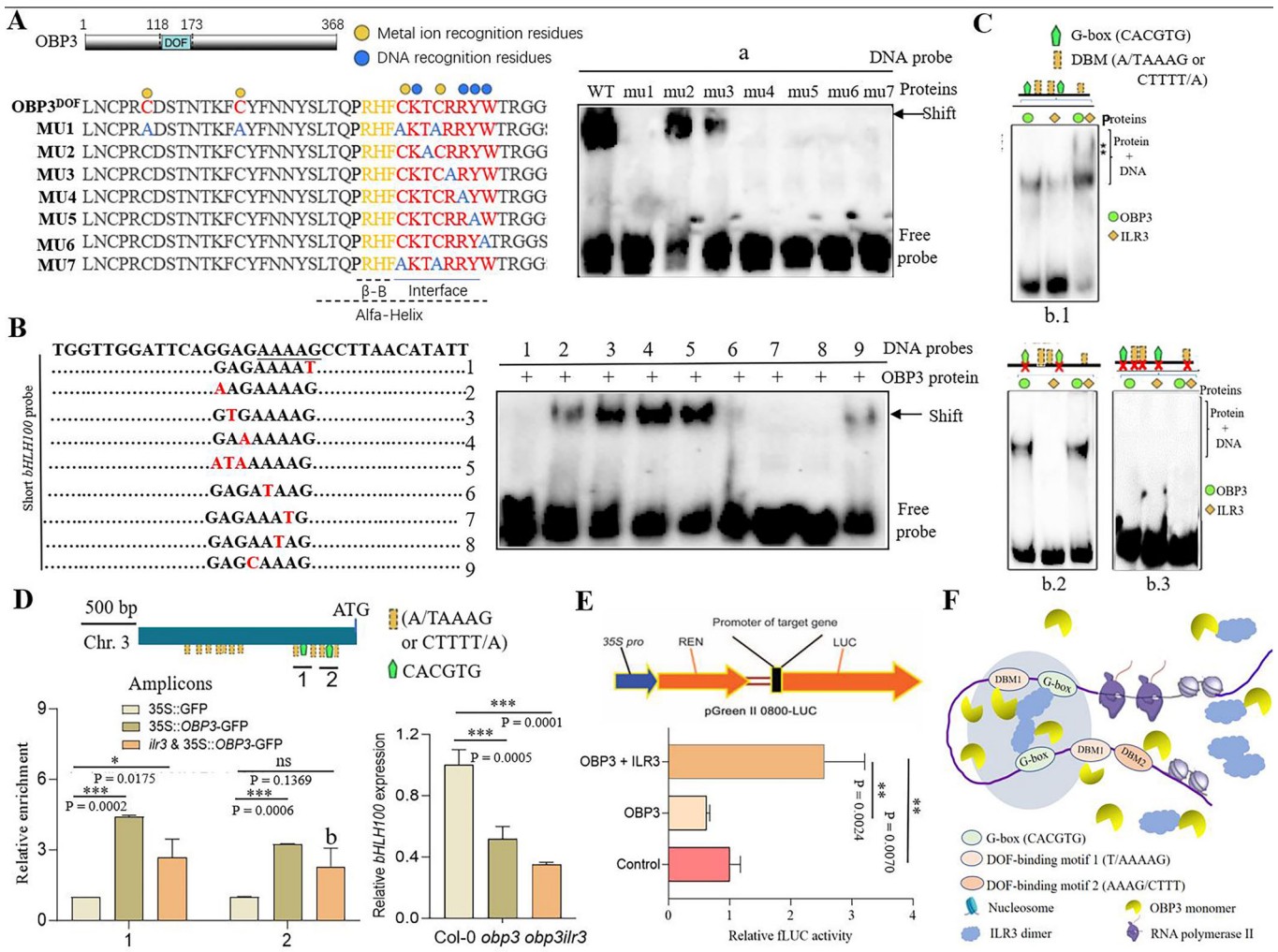

**Figure 6. Molecular basis and cooperation of ILR3 and OBP3 binding to the *bHLH100* promoter.**

(A) Displays the OBP3 DOF domain and the alignment of OBP3$^{DOF}$ and OBP3 mu1-7 proteins in Arabidopsis. The EMSA analysis of mu1-7 forms of OBP3 proteins' ability to interact with the DNA probes is presented. (B) The interactions between OBP3$^{DOF}$ and wide-type and mutant DNA probes were assayed by EMSA. Experiments were performed twice independently, yielding similar results. (C) Gel-shift analysis was conducted to examine the interactions between single ILR3$^{bHLH}$, OBP3$^{DOF}$, or a combination of ILR3$^{bHLH}$ and OBP3$^{DOF}$ proteins with wild-type or mutant DNA probes. The stars indicate the super shift band. (D) The binding profile of OBP3 and ILR3 to the *bHLH100* gene was also analyzed. The DOF-motifs and G-boxes are shown throughout the gene promoter, and the locations of amplicons for ChIP-qPCR analysis are indicated. ChIP-qPCR analysis was performed on the transcribed regions of *bHLH100* in 35S::GFP and transgenic plants carrying 35S::OBP3-GFP in Col-0 and *ilr3* mutant backgrounds. qPCR analysis was conducted on *bHLH100* mRNA levels in Col-0, *obp3*, and *obp3ilr3* mutants' backgrounds. The data are presented as means ± SD, $n = 3$. Data were analyzed by two-tailed Student's $t$-test. Asterisk denotes significant differences (*$p < 0.05$, **$p < 0.01$, Student's $t$-test). (E) The effect of *ILR3* on *OBP3* transcriptional activity in Col-0 protoplasts was also investigated. The error bars represent SD ($n = 3$). Data were analyzed by two-tailed Student's $t$-test. Asterisk denotes significant differences (*$p < 0.05$, **$p < 0.01$, ***$p < 0.001$; Student's $t$-test). Exact $P$ values are provided. (F) the proposed model for the role of the OBP3-ILR3 module in regulating the transcription of genes involved in iron homeostasis. Upon chromatin opening, the G-box and DOF motifs become accessible. ILR3 binds to the G-boxes, while OBP3 binds to the DOF motifs near the G-boxes. ILR3 and OBP3 interaction occur when they are both bound to DNA. ILR3 binding to the G-boxes enhances OBP3 binding, enabling it to bind to DNA and recruit OBP3 to target promoter. Source data are available online for this figure.

proteins exhibit nuclear colocalization. Interactions between BTS and OBP3 proteins occur and are dependent on RING domain, which acts as an E3 ligase. This was confirmed using a Y2H assay (Fig. 8A). The BTS RING domain facilitate its interaction with OBP3. To further verify the interaction between them, bimolecular fluorescence complementation (BiFC) experiments were also used. The assays were performed in *N. benthamiana* epidermal cells (Fig. 8B). The results of BiFC assay were confirmed through Co-IP. OBP3 and ILR3 co-immunoprecipitated with full-length BTS

(Fig. 8C), supporting the BiFC results. The direct interaction between OBP3 and BTS was also confirmed by in vitro pull-down assay (Appendix Fig. 8).

The interactions between BTS and OBP3 proteins through E3 ligase (RING) domain-mediated ubiquitination, followed by 26S proteasome-mediated degradation. Previous studies have shown that BTS and its rice orthologs OsHRZ1/OsHRZ2 have the capacity for ubiquitination (Kobayashi et al, 2013). As the HHE structural domains are necessary for iron binding, and deletion of

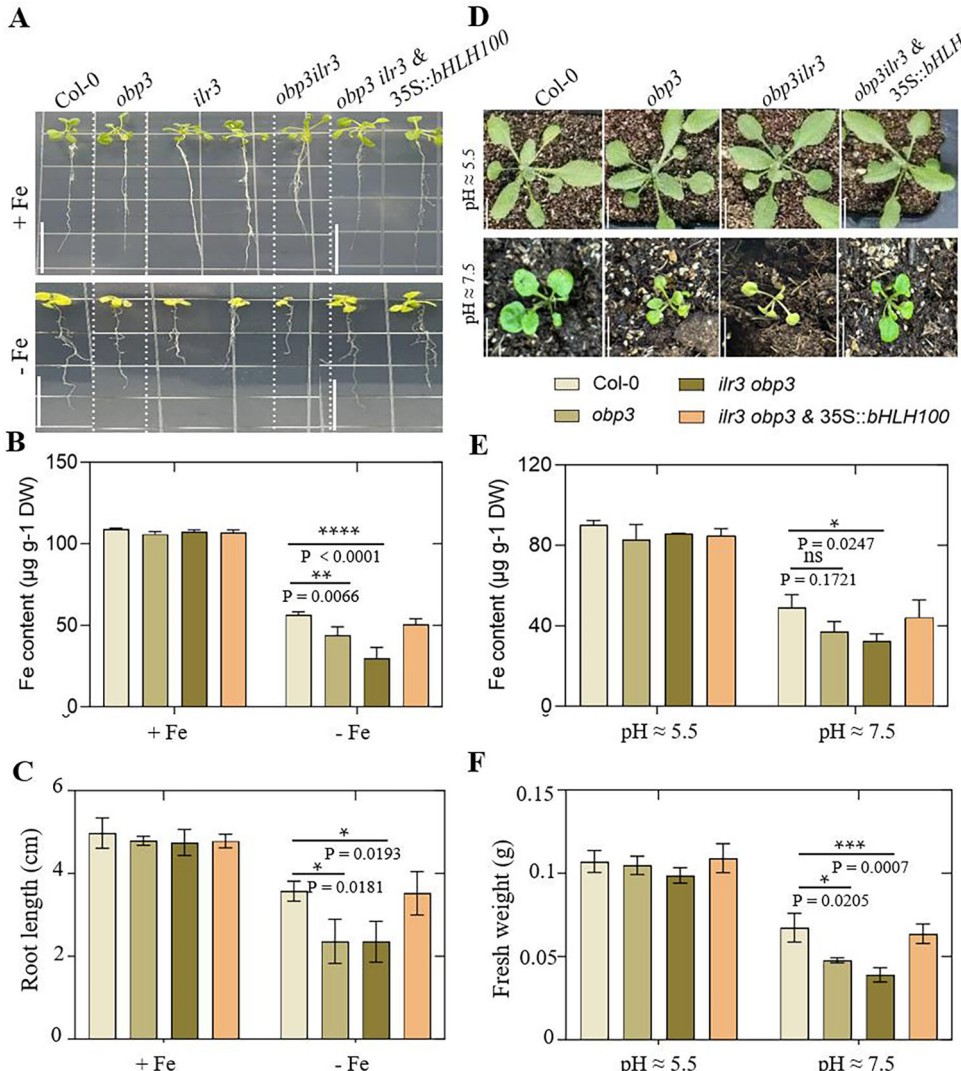

**Figure 7. Overexpression of *bHLH100* complements mutant's iron-deficient phenotype.**

(A) The phenotypes of Col-0, *obp3*, *ilr3*, *obp3ilr3* mutants, and *obp3ilr3* & 35S::*bHLH100* transgenic plants after 8 days of growth on Fe-sufficient (+ Fe, 50 μM Fe) or Fe-deficient (− Fe, 0.5 μM Fe) medium. The bar represents 2.0 cm. (B) The figure shows the Fe concentrations and (C) root length in plants grown on Fe-sufficient (+ Fe) or Fe-deficient (− Fe) medium for 8 days. DW, Dry weight. The data represent means ± SD ($n = 3$). Data were analyzed by two-tailed Student's *t*-test. Asterisk denotes significant differences (*$p < 0.05$, **$p < 0.01$, ****$p < 0.0001$; Student's *t*-test) (B, C). Exact $P$ values are provided. (D) Representative images of Col-0, *obp3*, *obp3ilr3* mutants, and *obp3ilr3* & 35S::*bHLH100* transgenic plants grown for 3 weeks on either normal soil (pH 5–6) or alkaline soil (pH 7–8) are shown. Chlorosis in mutant plants is observed. (E) Fe concentration and (F) fresh weight of rosette leaves were measured in Col-0, *obp3*, *obp3ilr3* mutants, *obp3ilr3* & 35S::*bHLH100* plants, $n = 3$. The bar indicates 1 cm. Results represent the means ± SD. Data were analyzed by two-tailed Student's *t*-test. Asterisk denotes significant differences (*$p < 0.05$, ***$p < 0.001$, ****$p < 0.0001$; Student's *t*-test) (E, F). Exact $P$ values are provided. Source data are available online for this figure.

these domains leads to an increased accumulation of truncated BTS, iron binding through these domains reduces the stability of BTS. Furthermore, the cell-free degradation assay was used to identify whether BTS is involved in the 26S proteasome-mediated degradation of OBP3 protein in vitro (Wang et al, 2009). ILR3 undergoes ubiquitinated degradation by BTS, while PYE proteins do not (Selote et al, 2015). Myc-tagged OBP3 protein was purified, quantified, and incubated with cell-free extracts prepared from Col-0, *bts-1*, and *bts-1*/proBTS::BTS$_{\triangle HHE}$-GFP seedlings grown under iron deficiency (Fig. 8D). Following iron deficiency treatment, we immunoblot analysis the impact of endogenous BTS on OBP3 protein content. The OBP3 protein appear to

degrade in Col-0 and *bts-1*/proBTS::BTS$_{\triangle HHE}$-GFP extract (Fig. 8D). To determine whether BTS-mediated ubiquitination triggers OBP3 degradation in vivo, we co-transfected BTS-GFP and Flag-OBP3 in Arabidopsis protoplasts. We observed a significant decrease in Flag-OBP3 protein abundance when in the presence of BTS-GFP; importantly, Flag-OBP3 degradation was recovered when protoplasts were treated with MG132 (a 26S proteasome inhibitor) (Appendix Fig. S9). These results suggest that BTS promotes the degradation of OBP3 via the 26S proteasome degradation pathway. To summarize, we propose a model for OBP3-mediated fine-tuning of plant homeostasis (Fig. 9).

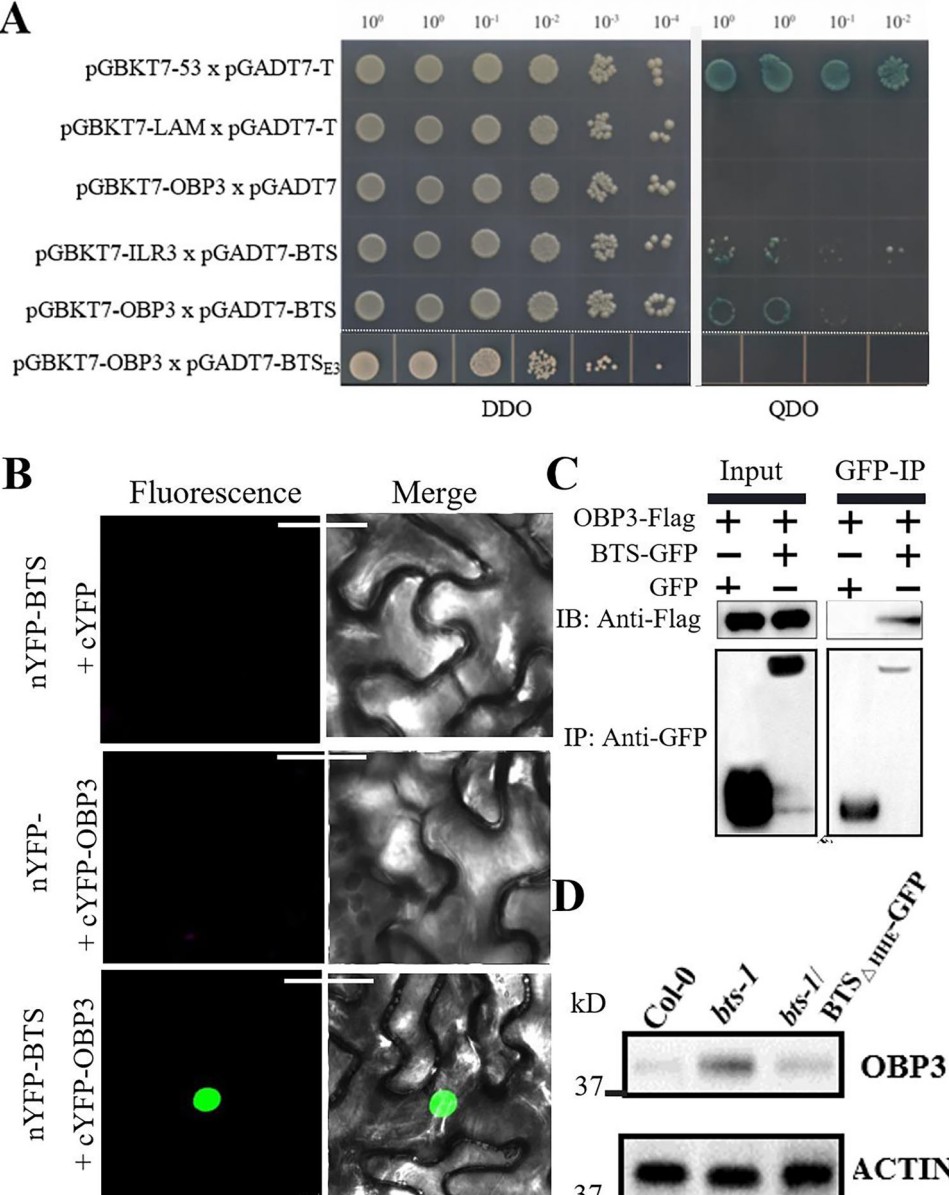

**Figure 8.   BRUTUS (BTS) targets OBP3 for degradation.**

(A) Y2H experiment was performed between BTS or BTS$_{\Delta E3}$ and ILR3, or OBP3 proteins. pGBKT7-53×pGBKT7-T is the positive control. pGBKT7-LAM×pGADT7-T is the empty control. The transformed yeast cells were selected on SD-Ade/-Trp/-Leu/-His medium. The plate was then spotted after diluting the yeast over a concentration gradient. (B) In planta BiFC assay was checked to investigate the OBP3 and BTS interaction. The various relevant constructs were co-transformed into *N. benthamiana*. Bar indicates 50 μm. (C) Co-IP assay was conducted. Protein was extracted from *N. benthamiana* leaves co-expressing OBP3-flag and BTS-GFP or OBP3-Flag and the empty GFP vector, and then immunoprecipitated by GFP antibody-conjugated agarose beads. The precipitated proteins and input samples were checked using anti-GFP or anti-Flag antibodies. (D) The degradation of OBP3 proteins by BTS and BTS$_{\Delta HHE}$ is mediated by the cell-free 26S proteasome. Myc-tagged OBP3 protein was expressed and purified. They were then incubated with cell-free protein extracts prepared from 7-d-old seedlings of Col-0, *bts-1*, and *bts-1*/proBTS::BTS$_{\Delta HHE}$-GFP grown on –Fe medium (4 d +Fe and 3 d –Fe). Proteins were detected using anti-Myc antibody. Actin is used as the western blot internal control. Source data are available online for this figure.

## Discussion

It had already been reported that the Ib subgroup *bHLH* genes played an important role in iron deficiency response in Arabidopsis (Sivitz et al, 2012). We used the *bHLH100* gene promoter as bait for yeast one-hybrid screening to identify transcription factors upstream of the gene. We obtained many positive colonies from the screening, seven of which bound to the promoter of *bHLH100*, which belongs to the known IVc subgroup bHLH genes. The other two positive clones were found to be the DOF family transcription factor OBP3. Further experiments confirmed that both OBP3 and ILR3 activated the *bHLH100* promoter (Fig. 1). OBP3 was further characterized by evaluating DNA- and protein-binding properties in vitro. In the studies, OBPs were expressed in *E. coli*. OBPs bind

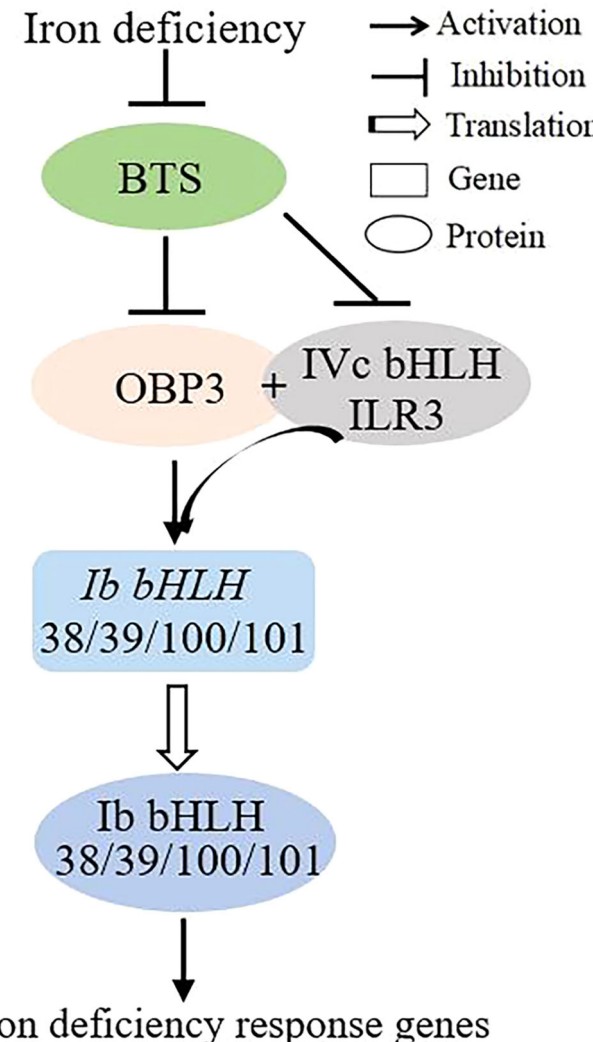

**Figure 9. The working model of OBP3 modulates iron homeostasis in Arabidopsis.**

The model illustrates the molecular function of OBP3 and ILR3 in the Fe homeostasis of Arabidopsis. OBP3 interacts with ILR3 to regulate the expression of Ib subgroup bHLH genes by binding to their promoters. OBP3 interacts with ILR3 (IAA-LEUCINE RESISTANT3), and their interaction enhances the DNA-binding ability and transcriptional promoting activity of OBP3, resulting in the positive regulation of iron deficiency-response genes. At the posttranscriptional level, the E3 ligase BRUTUS (BTS) aids in the degradation of the OBP3 and ILR3 proteins by the 26S proteasome machinery to prevent excessive iron uptake and maintain iron homeostasis in plants. Central regulatory proteins are represented by ovals and downstream target genes by squares. Source data are available online for this figure.

to the A/TAAAG element of *bHLH100* promoter, as shown in Fig. 1D. None of the OBPs bound to the mutant AAAG element, which contained a 2-bp mutation in each AAAG motif, indicating their specificity for that element (Fig. 6A,B). According to previous literature, OBPs can also bind to CTTT[T/A] elements from the Arabidopsis GST6 promoter in a similar manner (Kang et al, 2003; Kang and Singh, 2000).

The plant-specific zinc-finger protein family includes DOF proteins, which interact with specific bZIP proteins, High Mobility

Group (HMG) Proteins and other DOF proteins (Yanagisawa, 2002). The homology between OBP proteins is limited to the 52 amino acids DOF binding domain. Since this is the only common sequence, this domain may have multiple functions, being responsible for binding DNA and for protein–protein interactions. DNA-binding properties of OBP proteins are similar in vitro, and they can interact with the bZIP transcription factor OBF4, the results being consistent with the findings of Yanagisawa (1997) that DOF domains are essential for the DNA binding of maize DOF proteins and their interactions with HMG proteins (Yanagisawa, 2002). The zinc-finger domain has also been shown to be multifunctional in other proteins. The maize DOF1 protein *trans*-activation activity was inhibited in xanthogenic protoplasts, but not in green protoplasts, according to Yanagisawa (Yanagisawa, 2000). Maize PBF interacts with a bZIP protein Opaque2 (Zhan et al, 2018). The interaction between proteins also contributes to specificity, which can be further enhanced by tissue-specific expression patterns, resulting in different transcript profiles for different tissues. It has been also shown by Yanagisawa that the maize DOF1 protein can act as a *trans*-activator through multiple DOF-binding sites in maize C4 phosphoenolase promoters, whereas the maize DOF2 protein can function as a repressor of DOF1 *trans*-activation (Yanagisawa, 2000, 2002).

It was not observed that the OBP3 protein significantly *trans*-activated reporter genes with downstream gene promoters containing several DOF-binding motifs. There are several explanations for these results, such as the possibility that OBP3 did not act as a transcriptional activator but as a transcriptional inhibitor. However, a deletion of OBPs N-terminal fused to the *GAL4* DNA-binding domain was transcriptionally active, indicating that OBPs contained a *trans*-activation domain. We did not identify homology in these regions for any of the known activation domains like those observed in maize DOF1 (Cavalar et al, 2007). Transgenic plants overexpressing *OBP3* have previously shown that this protein plays an important role in plant development (Kang et al, 2003), although the true function of the gene remains unclear. In the current study, we have identified OBP3 as a key regulator of iron homeostasis (Fig. 9). To determine the role of *OBP3* in plants, we identified several functionally deficient *obp3* mutants. Genetic and phenotypic analyses of these *obp3* mutant lines have identified a key role of OBP3 in Fe homeostasis (Fig. 3). We further clarified that the balance between iron homeostasis and the accumulation of excess iron caused by *OBP3* overexpression was the major reason for the leaf chlorosis and dwarf phenotype of 35S::*OBP3* plants (Fig. 2). In addition, the mutant seedlings grew well in ½ MS medium with sufficient Fe content but showed abnormal growth in iron-deficient medium or alkaline soil with low Fe bioavailability, indicating that OBP3 was involved in iron hemostasis regulation. OBP3 bound specifically to the DOF recognition sequence [A/T] AAAG to activate transcription (Kang et al, 2003). We find that OBP3 positively regulates the key Fe homeostasis 1b subgroup *bHLH* genes and plant growth (Fig. 4).

We revealed here that plants regulate Fe homeostasis through co-regulation between OBP3 and the subgroup IVc bHLH ILR3, which strongly activate the subgroup Ib genes *bHLH38/39/100/101* transcription, a finding consistent with the previous observation that iron deficiency cascades signal precise layer-by-layer control of Fe assimilation under a Fe-deficient environment. The observation that the phenotypes of *obp3* and *ilr3* mutant roots are similar to those of the wild type under sufficient iron conditions, but are defective under low

iron conditions, suggests that these two regulatory genes play a specific role in the iron deficiency signaling pathway. OBP3 binding strength is increased in vivo when ILR3 is present, including *bHLH100* (Fig. 6D). Furthermore, in vitro, as a result of ILR3$^{bHLH}$ binding, OBP3$^{DOF}$ binds to a DNA fragment in which both DOF binding sites are mutated; this suggests that DNA allostery may enhance the affinity of OBP3$^{DOF}$ for the ILR3$^{bHLH}$–DNA complex. A TF recognizes its DNA-binding site by both direct interaction with specific base sequences and by recognition of local DNA features, such as DNA bending or unwinding. We propose that ILR3 enhanced DNA binding and altered the local shape of DNA to enhance OBP3 binding to DNA. Experiments need to be carried out to determine whether OBP3 affects ILR3 binding in vivo. It has been shown that the bHLH family TFs formed tetramers; these tetramers may facilitate DNA looping, as has been shown by PIF4 tetramers (Gao et al, 2022b), and may increase OBP3 binding in the vicinity. ILR3 and OBP3 proteins interaction potentially leads to the formation of a functional heterodimer, which accumulates in low iron conditions and facilitates uptake under iron deficiency.

To assess if BTS-mediated ubiquitination leads to OBP3 degradation, we co-transfected BTS-GFP and Flag-OBP3 in Arabidopsis protoplasts. Flag-OBP3 protein levels significantly decreased with BTS-GFP, but not its expression level. Treatment with 50 μM MG132, a 26S proteasome inhibitor, restored Flag-OBP3 levels (Appendix Fig. S9). This indicates that BTS1 facilitates OBP3 degradation through the 26S proteasome pathway in vivo. So, this study presents empirical evidence indicating that BTS specifically targets the 26S proteasome-mediated DOF transcription factor OBP3 and bHLH transcription factor ILR3, thereby impeding the formation of heterodimer complexes involving ILR3 and OBP3 proteins. Consequently, this interference may disrupt the regulation of downstream iron deficiency response genes controlled them. In addition, BTS also accumulates during iron deficiency conditions and potentially inhibits the OBP3/ILR3-mediated response. This fine-tuning of the iron deficiency response ensures appropriate iron absorption without excessive uptake, consequently impacting overall root growth. Overall, our results contribute to an improved understanding of how Fe deficiency regulates plant gene expression and how it contributes to normal growth.

# Methods

### Reagents and tools table

| Reagent/Resource | Reference or Source | Identifier or Catalog Number |
|---|---|---|
| **Experimental Models** | | |
| *obp3-1* | NASC | SALK_111615 |
| *obp3-2* | NASC | CS853401 |
| *bts-1* | NASC | SALK_099726 |
| *Ilr3-2* | NASC | SALK_016526 |
| **Recombinant DNA** | | |
| pMAL-c5X | NEB | HG-VYN0870 |
| pET30a(+) | Novagen | HG-VYN0173 |
| **Antibodies** | | |
| Anti-GFP | Roche | 11814460001 |

| Reagent/Resource | Reference or Source | Identifier or Catalog Number |
|---|---|---|
| Anti-Flag | MBL | M185-3L |
| Anti-MYC | Abclonal | AT070 |
| HRP Goat anti-Mouse IgG | Abclonal | AS003 |
| GFP-Nanoab-Agarose | Lablead | GNA-250-5K |
| Protein A/G Agarose | Yeasen | 36403es8 |
| **Oligonucleotides and other sequence-based reagents** | | |
| PCR primers | This study | Table EV3 |
| Chip-qPCR primers | This study | Table EV4 |
| EMSA probes | This study | Table EV5 |
| **Chemicals, Enzymes and other reagents** | | |
| Recombinant OBP3 protein | This study | Figure EV6 |
| β-estrogen | SIGMA | E2257-1MG |
| GUS | Yuanye Bio-Technology | R22612-100ml |
| **Software** | | |
| Graphpad prism 7 | https://www.graphpad.com/ | |
| ImageJv1.8.0.345 | https://imagej.ssjas.cn/ | |
| **Other** | | |
| ROCHE LightCycler® 96 | ROCHE | |

## Plant materials and growth conditions

The *ilr3-2* (SALK_004997), *bts-1* (SALK_016526), *obp3-1* (SALK_111615), and *obp3-2* (CS853401) mutants were obtained from Nottingham Arabidopsis Stock Centre (NASC; Nottingham, UK), and *ilr3obp3* double mutant was obtained by crossing. All mutants are in Col-0 background. Seeds were sterilized and vernalized at 4 °C for 4 days before germination at 20 °C. The Fe-sufficient medium consisted of half-strength Murashige and Skoog media with 1.0% (w/v) sucrose and 0.9% (w/v) agar, with 50 μM Fe-EDTA at pH 5.7. For Fe-deficient media, the same half-strength MS medium (without Fe) supplemented with 0.5 μM Fe-EDTA at pH 5.7. *A. thaliana* plants were grown at a temperature of 20 °C under long-day conditions, which consisted of a 16-h day and an 8-h night, with a relative humidity of 60%. Protoplasts were prepared from 3-week-old seedlings with rosette leaves. *N. benthamiana* were grown in climate-controlled growth chambers at 26 °C for 6 weeks with 16 h light and 8 h dark.

## Plasmid construction and transformation

We amplified the coding sequences of OBP3, ILR3, and BTS, along with the 3×HA, Flag, Myc, and GFP tag sequences, and cloned them sequentially into the pHB vector under CaMV 35S promoter or the pER8 vector with estrogen-inducible promoter, to obtain 35S::OBP3-Flag, 35S::*OBP3*-Myc (EQKLISEEDL), 35S::*OBP3*-GFP (Green Fluorescent Protein), 35S::BTS-GFP, and 35S::ILR3-GFP vectors. The recombinant plasmids were introduced into *Agrobacterium tumefaciens* strain GV1301. The primers used are listed in Appendix Table S3.

## RT-qPCR analysis

Total RNA was extracted from Arabidopsis seeds, following the RNeasy Plant Mini Kit (QIAGEN, Germany) manufacturer's instructions. Subsequently, cDNA was synthesized from 1 µg total RNA using Superscript II Reverse Transcriptase (Clontech, Japan). The experiment was conducted with a Light Cycler® 96 Real-Time PCR System (Roche, Germany). Each sample was subjected to three biological replicates. The primers used are listed in Appendix Table S4.

## Protoplast transcription activity assay

To analysis transcription activity, plasmids were constructed by cloning promoters of *OBP1-4*, *ILR3*, and *bHLH100* genes by PCR. These were then individually inserted into the 35S-SUC carrier Clontech, Japan). To detect the regulation by OBP3 on subgroup 1b bHLH promoters, effector plasmids (35S::GFP, OBP3::HA, or ILR3-GFP), reporter plasmids (probHLH100-LUC), and were introduced into Arabidopsis protoplasts. Luciferase activity was measured using the Dual-LUC Reporter Analysis System (Promega, USA). The bHLH100 promoters were contained in the pGreenII-0800 LUC construct, while the ILR3 coding region was present in the p62-SK construct. These constructs were co-transformed and the Firefly LUCIFERASE (LUC) and Renilla LUCIFERASE (REN) activities were measured. Relative activities of *OBP3* transcription were represented by the ProOBP3::LUC/Pro35S::REN ratio.

## Recombinant OBP3 protein purification and EMSA assay

Expression vector pMAL-C5x and pET30a (GE Healthcare, USA) were used to clone the *OBP3* and *ILR3* genes. The recombinant protein expression was induced using 0.2 mM IPTG (isopropyl β-D-1-thiogalactopyranoside) at 28 °C for 4 h, and purified by using Ni-NTA agarose (GE Healthcare). For EMSA assay, the reaction samples contained 1 µg of poly (dI:dC), 0.5 µg of recombinant His-OBP3 protein, and 10 fmol of biotin-labeled double-stranded DNAs. The samples were incubated at room temperature for 20 min and then subjected to electrophoresis with 0.5×TBE (Tris/borate/EDTA) buffer (pH 8.3) using the Lightshift Chemiluminescent EMSA Kit (Thermo Scientific, USA). Appendix Table S5 lists the primers and probes used.

## ChIP-qPCR experiment

Chromatin was isolated following the protocol described in Xu et al (2020), with chromatin from wild-type serving as the control. DNA prepared for ChIP was analyzed using SoFast™ qPCR Supermix (BioRad, USA) and a Light Cycler 96 Real-Time PCR Detection System (Roche). The percentage of target gene promoter enrichment was calculated relative to the input DNA, with the *ACTIN2* serving as a non-specific target. The primers used in qPCR analyses are listed in Appendix Table S6.

## Determination of ferric-chelate reductase activity

Ferric-chelate reductase activity was checked using spectrophotometry, following previously described method (Chen et al, 2010).

Ten plants of each strain were pretreated in a plastic container with 5 mL of 1/2 MS solution without micronutrients for 30 min (pH 5.5). Subsequently, they were soaked in 5 mL of Fe (III) reduction analysis solution [MS without trace elements containing 100 µM Fe (III)-EDTA and 300 µM ferrozine, adjusted to pH 5.0] for 30 min. The blank solution without plants was used as a control. The purple Fe (II)-Fe-Zn complex was quantified using a molar extinction coefficient of 28.6 mM$^{-1}$ cm$^{-1}$ at 562 nm.

## Analysis of iron concentration

For tissue Fe concentration analysis, plants were placed on Fe-sufficient or Fe-deficient MS medium, and approximately 30 mg of tissue was harvested. The tissue samples were dried at 80 °C for 24 h, digested with 10 mL of concentrated nitric acid: perchloric acid (3:1), and diluted to 10 mL in ddH$_2$O. Fe concentration was determined using inductively coupled plasma-atomic emission spectrometry.

## Perls' Prussian blue staining of iron

To localize Fe$^{3+}$, plant tissue was vacuum infiltrated with Perls' staining solution, which is a mixture of 4% (v/v) HCl and 4% (w/v) potassium ferricyanide for 20 min. Samples were then incubated in the staining solution for an additional 60 min and rinsed with ddH$_2$O five times. Plant tissue was incubated with the fixative solution (methanol:chloroform:acetic acid, 6:3:1) at room temperature for 2 h and then transferred back to Perls' staining solution. The localization of Fe$^{3+}$ was observed using light microscopy (Leica, Germany).

## Quantitative analysis of GUS activity

A volume of 100 µl of protein supernatant was combined with 400 µl of GUS extraction buffer that had been pre-warmed to 37 °C. The GUS extraction buffer involves the addition of 50 ml of a 0.1 M phosphate buffer solution with a pH of 7.0, 1 ml of a 10% SDS solution, 2 ml of a 0.5 M EDTA solution with a pH of 8.0, 100 µL of Triton X-100, 100 µL of β-mercaptoethanol, and finally, the addition of 100 ml of water. Subsequently, 500 µl of MUG substrate was added, and the mixture was incubated in a warm bath at 37 °C. At time points of 0 min, 15 min, 30 min, 45 min, and 60 min, 200 µl of the reaction mixture was combined with 800 µl of the reaction stop solution and stored in a dark room at room temperature. The fluorescence intensity values at different time points were measured using a fluorescence spectrophotometer with an excitation wavelength of 365 nm, an emission wavelength of 455 nm, and a slit width of 10 nm. The fluorescence intensity value serves as a representation of the reaction time, and the rate of change of fluorescence intensity over a specific time interval is determined. In addition, the rate of change of fluorescence intensity per unit mass of protein involved in the reaction is calculated by dividing the rate of change of fluorescence intensity per unit time by the quantity of protein.

## Chlorophyll concentration analysis

Chlorophyll was extracted from the seedlings and leaves using 80% acetone in 2.5 mM HEPES-KOH buffer (pH 7.5), and chlorophyll

content was checked according to the method described by Wellburn (Wellburn, 1994).

## Yeast one-hybrid assay

The Arabidopsis transcription factor (TF) library comprises 1589 TFs. It was constructed by OE Biotech (Shanghai, China) by cloning the Arabidopsis TF gene sequences into the pDEST22 vector and transforming Y187 competent yeast cells (Ou et al, 2011). To identify TFs that interact with the bait protein in the Y1H system, we amplified promoter region of the bait gene. Next, we integrated the bait gene into the chromosome of the bait yeast strain YM4271. The screening was carried out using the selective medium SD/-Leu/-His/3-AT (3-amino-1,2,4-triazole, 30 mM).

## Yeast two-hybrid assay

Y2H library was constructed by Wuhan BioRun Bioscience Co., Ltd., China. The screening method was described in a previous report by Osman (Osman, 2004). The OBP3 sequence was cloned into the pGBKT7 bait vector and transformed into the yeast strain AH109. The yeast cells carrying the bait vector were subsequently transformed with the prey plasmids containing bHLH115, bHLH121, bHLH39, ILR3, and PYE. The transformants were selected on SD-Trp/-Leu/-His medium and SD-Ade/-Trp/-Leu/-His medium.

## BiFC assay

The pSY728 and pSY738 vectors were used to clone the full-length OBP3 and ILR3 DNAs, respectively, resulting in fusions with the N- and C-terminal halves of Yellow Fluorescent Protein (YFPn and YFPc). A. tumefaciens strain GV3101 was used to transform tabacum leaves. Two days after transformation, the leaves were observed using a confocal microscope (Leica SP8, Germany).

## Co-immunoprecipitation assay

A. tumefaciens strain GV3101, which contained 35S::ILR3-GFP expression vector, was infiltrated together with 35S::OBP3-myc Agrobacterium into the tobacco leaves. The transformed tobacco leaves were collected for the co-immunoprecipitation assay 2 days after infiltration, as previously described (Xu et al, 2020; Gao et al, 2020). The ILR3-GFP precipitated by OBP3-myc was detected using an antibody that recognizes GFP (Abmart, China), while OBP3-myc was detected using an antibody that recognizes myc (Abmart).

## GST pull-down assay

The GST-BTS and His-OBP3 fusion proteins were purified using glutathione and Ni-NTA agarose beads, respectively. 2 μg of BTS-GST-bound beads were incubated with 2 μg HIS-OBP3 in binding buffer (20 mM Tris-HCl, pH 7.5, 200 mM NaCl) at 4 °C for 2 h. The beads were washed three times with the same buffer. The bound proteins were eluted, boiled in 50 μL of 2× sampling buffer, and analyzed by 12% SDS-PAGE and western blot using anti-HIS or anti-GST antibodies.

## Cell-free degradation assay

The experiment was carried out using protein extracts from 7-d-old wild-type, *bts-1*, and *bts-1*/proBTS::BTS$_{\triangle HHE}$-GFP seedlings as previously reported (Wang et al, 2009). Myc-tagged bHLH transcription factor ILR3 and OBP3 were translated and purified in vitro using anti-Myc affinity resin. The assay for degradation mediated by the 26S proteasome involved incubating proteins Myc-tagged ILR3 and OBP3 with cell-free protein extracts. The reactions were carried out at 22 °C for 8 h, followed by separation on a 10% (w/v) SDS-PAGE gel. Immunodetection was conducted.

## Statistical analysis

We utilized two-tailed Student's *t*-test with Office Excel software analysis for comparison of two samples.

# Data availability

This study includes no data deposited in external repositories.

The source data of this paper are collected in the following database record: biostudies:S-SCDT-10_1038-S44318-024-00304-0.

# Peer review information

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

## Acknowledgements

We thank NASC Arabidopsis seed stock center for providing the T-DNA mutants. This work was supported by the National Natural Science Foundation of China (Grant No. 32371293, No. 32171232, No. 31570859, No. 31500236, and No. 90917009) and Natural Science Foundation of Shanghai (Grant No. 22ZR1469500). We appreciate the funding from China's manned space program.

## Author contributions

**Peipei Xu**: Conceptualization; Formal analysis; Supervision; Funding acquisition; Investigation; Methodology; Writing—review and editing. **Yilin Yang**: Resources; Software; Validation. **Zhongtian Zhao**: Resources; Software; Methodology. **Jinbo Hu**: Software; Investigation; Methodology. **Junyan Xie**: Conceptualization; Formal analysis. **Lihua Wang**: Visualization; Project administration. **Huiqiong Zheng**: Software; Formal analysis; Supervision. **Weiming Cai**: Conceptualization; Formal analysis; Funding acquisition; Investigation.

Source data underlying figure panels in this paper may have individual authorship assigned. Where available, figure panel/source data authorship is listed in the following database record: biostudies:S-SCDT-10_1038-S44318-024-00304-0.

## Disclosure and competing interests statement

The authors declare no competing interests.

