## [Peer Review File · The EMBO Journal]

The transcription factor Dof3.6/OBP3 regulates iron homeostasis in Arabidopsis

peipei Xu, yilin yang, zhongtian zhao, jinbo hu, Junyan Xie, lihua wang, Hui Qiong Zheng, and Weiming Cai

Corresponding author(s): peipei Xu (ppxu@cemps.ac.cn) , Hui Qiong Zheng (hqzheng@cemps.ac.cn), Weiming Cai (wmcai@cemps.ac.cn)

Review Timeline:

Submission Date:	17th Jan 24
Editorial Decision:	21st Feb 24
Revision Received:	27th Jul 24
Editorial Decision:	16th Sep 24
Revision Received:	19th Oct 24
Accepted:	24th Oct 24

Editor: William Teale

Transaction Report:

Dear Dr Xu,

Thank you again for the submission of your manuscript entitled "Dof3.6/OBP3, a DOF transcription factor gene, modulates iron homeostasis in Arabidopsis". We have now received the reports from the referees, which I copy below.

As you can see from their comments, the referees state that the topic is interesting and the contribution that this manuscript represents is significant and timely. That said, all of them point out that a significant amount of further work that will require your attention before your manuscript can be published in The EMBO Journal.

Based on the overall interest expressed in the reports, I would like to invite you to address the comments of all referees in a revised version of the manuscript. I should add that it is The EMBO Journal policy to allow only a single major round of revision and that it is therefore important to resolve the main concerns at this stage. I believe the concerns of the referees are reasonable and addressable, but please contact me if you have any questions, need further input on the referee comments or if you anticipate any problems in addressing any of their points. I recommend we talk via Zoom once you have had a chance to digest the referee reports. Please, follow the instructions below when preparing your manuscript for resubmission.

I would also like to point out that as a matter of policy, competing manuscripts published during this period will not be taken into consideration in our assessment of the novelty presented by your study ("scooping" protection). We have extended this 'scooping protection policy' beyond the usual 3 month revision timeline to cover the period required for a full revision to address the essential experimental issues. Please contact me if you see a paper with related content published elsewhere to discuss the appropriate course of action.

Again, please contact me at any time during revision if you need any help or have further questions.

Thank you very much again for the opportunity to consider your work for publication. I look forward to your revision.

Best regards,

William

William Teale, Ph.D.
Editor
The EMBO Journal

When submitting your revised manuscript, please carefully review the instructions below and include the following items:

- 1) a .docx formatted version of the manuscript text (including legends for main figures, EV figures and tables). Please make sure that the changes are highlighted to be clearly visible.
- 2) individual production quality figure files as .eps, .tif, .jpg (one file per figure).
- 3) a .docx formatted letter INCLUDING the reviewers' reports and your detailed point-by-point response to their comments. As part of the EMBO Press transparent editorial process, the point-by-point response is part of the Review Process File (RPF), which will be published alongside your paper.
- 4) a complete author checklist, which you can download from our author guidelines ([https://wol-prod-cdn.literatumonline.com/pb-assets/embo-site/Author Checklist%20-%20EMBO%20J-1561436015657.xlsx](https://wol-prod-cdn.literatumonline.com/pb-assets/embo-site/Author%20Checklist%20-%20EMBO%20J-1561436015657.xlsx)). Please insert information in the checklist that is also reflected in the manuscript. The completed author checklist will also be part of the RPF.
- 5) Please note that all corresponding authors are required to supply an ORCID ID for their name upon submission of a revised manuscript.
- 6) We require a 'Data Availability' section after the Materials and Methods. Before submitting your revision, primary datasets produced in this study need to be deposited in an appropriate public database, and the accession numbers and database listed under 'Data Availability'. Please remember to provide a reviewer password if the datasets are not yet public (see <https://www.embopress.org/page/journal/14602075/authorguide#datadeposition>). If no data deposition in external databases is

needed for this paper, please then state in this section: This study includes no data deposited in external repositories. Note that the Data Availability Section is restricted to new primary data that are part of this study.

Note - All links should resolve to a page where the data can be accessed.

8) For data quantification: please specify the name of the statistical test used to generate error bars and P values, the number (n) of independent experiments (specify technical or biological replicates) underlying each data point and the test used to calculate p-values in each figure legend. The figure legends should contain a basic description of n, P and the test applied. Graphs must include a description of the bars and the error bars (s.d., s.e.m.).

9) We would also encourage you to include the source data for figure panels that show essential data. Numerical data can be provided as individual .xls or .csv files (including a tab describing the data). For 'blots' or microscopy, uncropped images should be submitted (using a zip archive or a single pdf per main figure if multiple images need to be supplied for one panel). Additional information on source data and instruction on how to label the files are available at .

10) We replaced Supplementary Information with Expanded View (EV) Figures and Tables that are collapsible/expandable online (see examples in <https://www.embopress.org/doi/10.15252/embj.201695874>). A maximum of 5 EV Figures can be typeset. EV Figures should be cited as 'Figure EV1, Figure EV2" etc. in the text and their respective legends should be included in the main text after the legends of regular figures.

12) Our journal encourages inclusion of *data citations in the reference list* to directly cite datasets that were re-used and obtained from public databases. Data citations in the article text are distinct from normal bibliographical citations and should directly link to the database records from which the data can be accessed. In the main text, data citations are formatted as follows: "Data ref: Smith et al, 2001" or "Data ref: NCBI Sequence Read Archive PRJNA342805, 2017". In the Reference list, data citations must be labeled with "[DATASET]". A data reference must provide the database name, accession number/identifiers and a resolvable link to the landing page from which the data can be accessed at the end of the reference. Further instructions are available at .

Further instructions for preparing your revised manuscript:

We realize that it is difficult to revise to a specific deadline. In the interest of protecting the conceptual advance provided by the work, we recommend a revision within 3 months (21st May 2024). Please discuss the revision progress ahead of this time with the editor if you require more time to complete the revisions. Use the link below to submit your revision:

Referee #1:

The manuscript by Xu et al shows that OBP3 modulates Iron (Fe) homeostasis in Arabidopsis. Overall, manuscript is well written, authors have done a decent job by doing many experiments to support their findings and conclusions. However, I have a number of major questions/comments regarding the data presented and there are still many issues that authors should take into account.

Major comments:

- Authors have shown excess accumulation of Fe in OBP3 expression lines based on Perls staining (Fig.2 C). I am surprised there is no stain at all in Col-0 leaf. Is this correct? Is there no Fe at all in WT plants? Authors need to be very specific which 35S:OBP3 line is used in Figure as well as how many replicates/leaves were imaged etc.
- FRO2 is a direct downstream target of bHLH38, bHLH39 and both of these genes are downregulated in the *obp3* mutant. So, the FRO2 expression is also expected to be less in the *obp3* mutant which is expected to affect the overall FRO2 activity. So, what happens to the expression of FRO2 and IRT1 in the *obp3* mutant?? As the mutant has less iron and it is more sensitive to -Fe so it is expected to have less FRO2 and IRT1 expression.
- In Fig.2 C the phenotyping data is not clear how many roots/individuals were counted? I could even see that *obp3* mutants root length is bit smaller than WT plants even under +Fe conditions? Why not complete details are given in figure legends. This seems consistent in all the figures. Many details are omitted and quality of plants images are not at par. Please given n=? Information in all the figures wherever you have presented quantified data. I assume the plates +Fe and -Fe are different scale car should be shown in both the conditions.
- Authors have shown Soil grown plant images at high pH conditions, with bleached true leaves. I think at high pH Fe on soil young leaves should show prominent chlorotic phenotype which I don't see in this plants. Like the standard norm in the field, authors should also phenotype this plants by supplemented external Fe (for example: Fe-EDDHA) at high pH which will rescue the chlorotic phenotype to make sure whatever you see is due to Fe limiting condition at high pH.
- Figure 4 B authors have shown the direct regulation of OBP3 on 1b subgroup of BHLH genes using EMSA/qRT-PCR and ChIP-qPCR. It would be important to show the expression of all 1b subgroup genes in *obp3* mutant plants in addition to 35S:OBP3 plants. Authors have shown the expression of only bHLH39 and bHLH100 (in Supple fig. 2). I wonder why they haven't shown the expression of other two genes in mutant lines.
- Authors have shown OBP3 acts upstream of 1B subgroup of genes and concluded that OBP3 is essential for bhlh38, 39, 100, 101 expression. So, it is also important to check what happens to the expression of genes like IRT1 and FRO2, as these genes are regulated by these BHLH TFs.

- In the Y2H screen authors have identified several putative prey proteins by using OBP3 as bait. Why not show all the proteins from your screen. I could see the in the table not many proteins are shown. Please show your results unbiased way. I could see the list is not consistent as many sr. no. is missing in between.
- In figure 8 (C), there is band in only GFP lane also which is a negative control so how can it be concluded that OBP3 is pulled down with BTS-GFP ??
- In luciferase assay (Fig. 1H), why OBP3 required to fuse with VP16 activation domain. The authors given an analogy. It contains both the repression and activation domains, so OBP3 cannot induce the expression of bHLH100.
- However, when authors check the expression of Ferritin genes by using 35S:OBP3 line, full-length OBP3 protein induces the expression of FER1 and FER2 (Supplementary Fig. 3 A). What will be a possible explanation? If the author's analogy is correct, please show the expression of bHLH100 in the 35S:OBP3 line is also not affected.
- Also, In transient experiments, the author used the activation domain of bacterial proteins. The authors should show the relative expression of all the tested genes in the 35:OBP3 or XVE:OBP3 line, shown in the luciferase assay.
- For the phenotyping with the XVE:OBP3 line, the author used 30 μ M β -estrogen which itself causes severe stress for plants. Please show the WT plants.
- Why do authors show phenotype for over-expression line in shoot and for mutant in the root? For consistency, author can show the phenotype of both root and shoot of mutant and over-expression line.
- Over-expression of bHLH100 in obp3 ilr3 double mutant background (35S:bHLH100 (obp/ilr3) rescued the root growth phenotype under -Fe conditions. What was the observation of 35S:bHLH100 (wt) phenotype?
- A single mutant of obp3, ilr3, and its double mutant obp3ilr3 have the same primary root length phenotype. How does OBP3 and ILR3 interaction regulate the primary root growth and what is the significance of this interaction in root growth?

Minor comments:

Line 64: Check the typo, it should be bHLH100/ bHLH101

Line 66-68 It should be "subgroup Ib genes responsible for "activation of" iron uptake genes" and not" subgroup Ib genes responsible for iron uptake.

Line 130: Here authors have mentioned a transformation-based approach for Y1H. But inn methodology, mating-based yeast is mentioned. Please clarify. If mating based approach have used, please add plate images of screen as well.

Line 144: Grown should be replaced with transferred.

Line 148: OBP3 cDNA have used as prey, right? and the bHLH100 promoter as a bait. Please correct.

Line 166: Yeast one hybrid will verify the interaction only, not regulation. Right? Please correct.

Line 188: OBP3 mutant generated or confirmed? as it was ordered from the NASC. Please clarify.

Line 322: Dexamethasone will not induce OBP3 expression in the OBP3-GR line, it will change the localization of OBP3 from the cytoplasm to the nucleus.

Line 591-592: It is important to discuss how Fe uptake is affected because of this regulation.

Referee #2:

The manuscript titled "Dof3.6/OBP3, a DOF transcription factor gene, modulates iron homeostasis in Arabidopsis" by Hu et al. offers valuable insights into the regulatory mechanisms orchestrating iron homeostasis in Arabidopsis. It notably underscores the role of OBP3 as a transcription factor for iron-related genes and its interaction with the established iron sensor, BTS E3 ligase protein. The discovery of a novel role for OBP3 within the iron deficiency pathway significantly advances our understanding of the intricate network governing iron homeostasis. To further enhance the detailed comprehension of OBP3's functions, addressing the following critical concerns is essential to provide robust support for the conclusions:

1- Assess the Regulation of OBP3 Protein by BTS: Conduct experiments to determine dynamic changes in OBP3 protein abundance in response to varying iron concentrations. This would reinforce the observation that the OBP3 protein level is regulated by BTS.

2- Investigate the Presence of a BTS Interaction Domain (BID) in OBP3: Strengthen the evidence for the potential interaction between OBP3 and BTS by exploring the presence of a BTS Interaction Domain (BID) in OBP3. Additionally, perform ubiquitination assays or pull-down assays to provide further support for the direct regulation of OBP3 by BTS.

3- Validate OBP3's Regulatory Role on Transcription Factors: The study highlights OBP3's regulatory role on iron deficiency-induced transcription factors, including bHLH38/39/100/101, through binding to gene promoter regions with ILR3 protein. To validate this claim, examine the mRNA levels of these transcription factors, not just bHLH100, in the *obp3* and *ilr3* double mutant. This experiment would elucidate whether OBP3 and ILR3 work synergistically to regulate the expression of these downstream targets.

4- The pronounced phenotype exhibited by the *obp3* mutant lines, particularly under iron-limiting conditions, warrants specific attention/discussion. Of note, a recent publication in *Nature Plants* (2023) has delineated the interconnected impact of iron content on plant growth, emphasizing that this relationship operates through an independent pathway (Zheng, Leiyong, et al. "The SOD7/DPA4-GIF1 module coordinates organ growth and iron uptake in Arabidopsis." *Nature Plants* 9.8 (2023): 1318-1332). Moreover, BioGrid data implies a potential interaction between OBP3 and GRF3 (<https://thebiogrid.org/10019/summary/arabidopsis-thaliana/obp3.html>). Significantly, mutations in GRF3 are correlated with diminished leaf size, suggesting its involvement in leaf development. This prompts an inquiry into whether the observed growth defects in the *obp3* mutant lines in this study are a direct consequence of altered iron content or are mediated through an independent pathway.

In conclusion, while the study significantly advances our understanding of iron homeostasis in *Arabidopsis*, the proposed experiments are recommended for enhanced clarity and further support the claims. By addressing these concerns, the manuscript's impact will undoubtedly be elevated, emphasizing the pivotal role of OBP3 in regulating iron homeostasis in *Arabidopsis* and providing valuable avenues for future research directions.

Referee #3:

In this manuscript, Xu et al. identified the DOF transcription factor OBP3 (AtDof3.6) in a yeast one-hybrid screen using the promoter of the *Arabidopsis* bHLH100 transcription factor as a bait. bHLH100 is a member of the bHLH clade Ib, which is known to regulate the iron deficiency response together with three other members of this clade, namely bHLH38, bHLH39 and bHLH101. In this study, the authors further characterize the role of OBP3 in regulating the iron deficiency response in *Arabidopsis* using gain- and loss-of-function mutants. In addition, the authors show that OBP3 interacts with ILR3, another key bHLH transcription factor involved in this process, to regulate the expression of bHLH100 and iron uptake. At the post-transcriptional level, the E3 ligase BRUTUS (BTS) assists in the degradation of the OBP3 protein by the 26S proteasome to prevent excessive iron uptake and maintain iron homeostasis in plants. The findings of this manuscript are novel and of interest to researchers on plant iron signalling and iron homeostasis regulation, the logic is reasonable, and the experiments are well performed. Therefore, this manuscript should be suitable for publication in this journal. However, I have some minor commendations listed below on this manuscript that need to be addressed before acceptance.

1. Please provide a reference if it is known that the promoter of bHLH100 is 1541 bp long.

2. In Figure 1C, please indicate the negative control in the legend of the figure.

3. The difference in GUS activity between +Fe and -Fe is limited and difficult to appreciate. The text is grammatically correct and follows a clear and concise structure. To show quantitative differences in promoter activity, quantitative GUS analysis is usually conducted.

4. To provide a complete understanding of OBP3 activity on this clade of bHLH transcription factors, it is recommended to include expression data for the two additional genes including bHLH38 and bHLH101 in the manuscript.

5. There is no confirmation of an increase in FERRITINS as no western blot was conducted. Therefore, the authors should state that expression analysis indicates a role for FERRITINS in protecting against oxidative damage.

6. What do FRO2 and IRT1 refer to?

7. Lines 76 and 86 should be defined.

8. In Figure 6A, what is the sequence of the DNA fragment used for the EMSA experiments?

9. The growth conditions, including light intensity, and temperatures, are incomplete.

Dear Editor and Reviewers,

We have resubmitted the manuscript entitled "*Dof3.6/OBP3, a DOF transcription factor gene, modulates iron homeostasis in Arabidopsis*" that was previously reviewed by the *EMBO Journal*.

We appreciate your letter and the insightful comments of the reviewers. These comments have greatly contributed to the improvement of our manuscript. We have responded to each reviewer's request in detail. In particular, we have also provided additional evidence to support the conclusion that *OBP3* regulates iron homeostasis. We hope that the revised version is suitable for publication in the *EMBO Journal*.

Yours sincerely

Peipei Xu

Referee #1:

The manuscript by Xu et al shows that *OBP3* modulates Iron (Fe) homeostasis in Arabidopsis. Overall, manuscript is well written, authors have done a decent job by doing many experiments to support their findings and conclusions. However, I have a number of major questions/comments regarding the data presented and there are still many issues that authors should take into account.

Response: We would like to thank you for your careful review of our manuscript and your recognition of the importance of our research. Your constructive suggestions have been duly considered, resulting in comprehensive modifications and revisions. We sincerely hope that the revised version of the manuscript will meet your expectations.

Major comments:

Authors have shown excess accumulation of Fe in *OBP3* expression lines based on Perls staining (Fig. 2C). I am surprised there is no stain at all in Col-0 leaf. Is this correct? Is there no Fe at all in WT plants? Authors need to be very specific which 35S::*OBP3* line is used in Figure as well as how many replicates/leaves were imaged etc.

Response: Thank you for your inquiry. We examined three independent 35S::*OBP3* transgenic lines, each with at least three rosette leaves. In fact, we can detect weak Perls' Fe staining signals in the leaves and epidermal hairs of wild-type Col-0 plants, while all transgenic plants overexpressing *OBP3* exhibit a strong staining signal in these tissues. A typical picture of the staining results is shown in the revised figure 2C. This may be attributed to the relative lower sensitivity of the

staining method compared to quantitative analysis method. This may not imply the absence of iron in wild-type Col-0 seedlings, as quantitative analysis confirms the presence of iron in the Col-0 seedlings (Figure 2D, 3D).

FRO2 is a direct downstream target of bHLH38, bHLH39 and both of these genes are downregulated in the *obp3* mutant. So, the FRO2 expression is also expected to be less in the *obp3* mutant which is expected to affect the overall FRO2 activity. So, what happens to the expression of FRO2 and IRT1 in the *obp3* mutant? As the mutant has less iron and it is more sensitive to -Fe so it is expected to have less FRO2 and IRT1 expression.

Response: Per your advice, we have performed the experiment to check the expression of *FRO2* and *IRT1* in the *obp3* mutants. Consistent with your speculation, their expression levels are lower in the *obp3* mutants. The results are shown below and in supplementary figure 4. Furthermore, we re-examined the iron reductase activity in wild-type and *obp3* mutant roots, and the results showed that the iron reductase activity decreased about 20 % to a significant level in the *obp3* mutant roots under half strength MS medium, this result supports the expression level data.

RT-qPCR analysis indicated *FRO2* and *IRT1* genes expression levels under Col-0 and the *obp3* mutants' background. The error bar represents SD. Star indicates significant difference by students' *t*-test.

Detection of Ferric-chelate reductase activity of the wild type and *obp3* mutants' roots. Ferric-chelate reductase activity was checked using spectrophotometry, following previously described method (Chen et al. 2010).The ferric-chelate reductase in Col-0 plants is set to 1. Values are means \pm SD. Significant differences from the wild type are indicated by an asterisk ($p < 0.05$), as determined by Student's *t*-test.

Chen WW, Yang JL, Qin C, Jin CW, Mo JH, Ye T, Zheng SJ (2010) Nitric oxide acts downstream of auxin to trigger root ferric-chelate reductase activity in response to iron deficiency in Arabidopsis. *Plant Physiol* 154 (2):810-819.

In Fig. 3C the phenotyping data is not clear how many roots/individuals were counted? I could even see that *obp3* mutants root length is bit smaller than WT plants even under +Fe conditions? Why not complete details are given in figure legends. This seems consistent in all the figures. Many details are omitted and quality of plants images are not at par. Please given n = ? Information in all the figures wherever you have presented quantified data. I assume the plates +Fe and -Fe are different scale car should be shown in both the conditions.

Response: Following your advice, we re-analyzed the root phenotypes of wild-type and mutant seedlings from the same growth stage on iron-sufficient and iron-deficient media, and selected at least 10 seedlings from each ecotype for statistical analysis of root length, the results showed that the root length of the mutant was not significantly different from that of the wild type on the iron-sufficient medium, but significantly shorter on the iron-deficient medium than that of the wild type control (Figure 3C, 3E). As per your suggestions, we've included detailed information, such as sample numbers and revised plant figures and bars, in the legends of Figures 3C & 3G, 2B & 2E, and 7A & B, etc. We hope this will meet with your requirements.

Authors have shown Soil grown plant images at high pH conditions, with bleached true leaves. I think at high pH Fe on soil young leaves should show prominent chlorotic phenotype which I don't see in these plants. Like the standard norm in the field, authors should also phenotype these plants by supplemented external Fe (for example: Fe-EDDHA) at high pH which will rescue the chlorotic phenotype to make sure whatever you see is due to Fe limiting condition at high pH.

Response: Following your suggestions, we have redesigned and carried out experiments on soil-grown plants under high pH conditions as described below. The results indicate that the *obp3* mutant exhibits a distinct chlorotic color phenotype when exposed to high pH soils. Furthermore, the introduction of external iron (100 μ M Fe-EDDHA) at the same time as the high pH treatment effectively rescued the chlorotic phenotype observed in the leaves of the *obp3* mutant. These results provide further evidence that the chlorotic phenotype of *obp3* leaves under high pH is caused by iron deficiency.

The phenotypes of three-week-old Col-0 wild type and *obp3* mutants grown on either alkaline soil (pH 7 to 8) or alkaline soil (pH 7 to 8) supplemented with Fe-EDDHA. Bar represents 1.0 cm.

Figure 4 B authors have shown the direct regulation of OBP3 on 1b subgroup of bHLH genes using EMSA/qRT-PCR and ChIP-qPCR. It would be important to show the expression of all 1b subgroup genes in *obp3* mutant plants in addition to 35S::*OBP3* plants. Authors have shown the expression of only bHLH39 and bHLH100 (in Supple fig. 2). I wonder why they haven't shown the expression of other two genes in mutant lines.

Response: Following your recommendation, we conducted an experiment to demonstrate the expression of the remaining two genes, *bHLH38* and *bHLH101*, in the mutant lines. The findings indicate that all 1b subgroup bHLH genes, including *bHLH38* and *bHLH101*, are downregulated in the *obp3* mutant background. The results are presented as below and have been incorporated into the revised supplementary figure 2B.

In the background of *obp3-2* mutant, iron deficiency induced upregulation of *bHLH38*, *bHLH39*, *bHLH100*, and *bHLH101* gene expression. The error bar represents SD. Star indicates significant difference by student's *t*-test.

Authors have shown OBP3 acts upstream of 1B subgroup of genes and concluded that OBP3 is essential for bHLH38, 39, 100, 101 expressions. So, it is also important to check what happens to the expression of genes like IRT1 and FRO2, as these genes are regulated by these bHLH TFs.

Response: Per your advice, we have performed the experiment to check the expression of *FRO2* and *IRT1* in the *obp3* mutants. Consistent with your speculation, their expression levels are lower in the *obp3* mutants. The results are shown below and in supplementary figure 4.

RT-qPCR analysis *FRO2* and *IRT1* genes expression levels under the *obp3* mutants' background. Star indicates significant difference by students' *t*-test.

In the Y2H screen authors have identified several putative prey proteins by using OBP3 as bait. Why not show all the proteins from your screen. I could see the in the table not many proteins are shown. Please show your results unbiased way. I could see the list is not consistent as many sr. no. is missing in between.

Response: Following your advice, we have made a comprehensive display of all potential interacting proteins screened by yeast two-hybrid. See the revised Supplementary Table. 2 for details.

In figure 8 (C), there is band in only GFP lane also which is a negative control so how can it be concluded that OBP3 is pulled down with BTS-GFP?

Response: Following a comprehensive examination, it was determined that the erroneous outcome was attributable to our failure to correctly identify the relevant lane. The original WB detection diagram is presented below, accompanied by the source data. Furthermore, the *in vitro* pull-down experiment was conducted, the results of which are shown below and in Supplementary Figure 7. These results further confirm the direct interaction between OBP3 and BTS both *in vivo* and *in vitro*.

(C) Co-IP assay was conducted. Protein was extracted from *N. benthamiana* leaves co-expressing OBP3-flag and BTS-GFP or OBP3-Flag and the empty GFP vector, and then immunoprecipitated by GFP antibody-conjugated agarose beads. The precipitated proteins and input samples were checked using anti-GFP or anti-Flag antibodies.

Supplemental Figure 8: The pull-down assay was conducted to verify the direct interaction between BTS and OBP3. The GST-BTS fusion protein and His-OBP3 fusion protein were purified using glutathione beads (GE Healthcare) and Ni-NTA agarose (GE Healthcare). 2 µg samples of BTS-GST-bound glutathione beads were incubated with 2 µg His-OBP3 in binding buffer containing 20 mM Tris-HCl, pH7.5 and 200 mM NaCl at 4°C for 2 h. Wash the beads three times with washing buffers (20mM Tris-HCl, pH7.5 and 200mM NaCl). Protein eluted from beads was loaded onto 12% SDS-PAGE gel by boiling in a 2 × sampling buffer of 50 µL and eluted by anti-HIS (GenScript, A00186) or anti-GST (GenScript, A00865-100) for WB analysis.

In luciferase assay (Fig. 1H), why OBP3 required to fuse with VP16 activation domain. The authors given an analogy. It contains both the repression and activation domains, so OBP3 cannot induce the expression of bHLH100. However, when authors check the expression of Ferritin genes by using 35S::OBP3 line, full-length OBP3 protein induces the expression of FER1 and FER2 (Supplementary Fig. 3 A). What will be a possible explanation? If the author's analogy is correct, please show the expression of bHLH100 in the 35S::OBP3 line is also not affected.

Response: Following your suggestion, we cancelled the fusion analysis of OBP3 protein and other DOF proteins with the VP16 activation domain. The results demonstrated that the OBP3 protein, when analyzed alone, was capable of activating the expression of the bHLH100 protein *in vivo*. The revised result is presented in Figure 1E. As shown in Figure 1E, the full-length OBP3 protein can still activate the expression of downstream genes, thereby upregulating the expression of the FER gene in the 35S::OBP3 transgenic plants. Furthermore, the result is consistent with the finding that the previous report that the *FER1/2* gene is induced by iron overload, while the expression of the bHLH1b subfamily genes, including *bHLH100*, is also upregulated in the 35S::OBP3 plants, as shown in Figure 4B.

Also, in transient experiments, the author used the activation domain of bacterial proteins. The authors should show the relative expression of all the tested genes in the 35S::OBP3 or XVE:OBP3 line, shown in the luciferase assay.

Response: Following your recommendation, we carried out an experiment evaluating the genes tested in the 35S::OBP3 lines, as shown in the figure below. The results showed that OBP3 overexpression does not affect the expression levels of other DOF genes. This is plausible because these DOF TFs have different molecular functions in Arabidopsis.

The relative *OBP1-4*, *DAG1*, *CDF1*, and *CDF2* gene expression levels in Col-0 and 35S::*OBP3* plants. The error bar represents SD. Star indicates significant difference by students' *t*-test.

For the phenotyping with the XVE: *OBP3* line, the author used 30 μ M β -estrogen which itself causes severe stress for plants. Please show the WT plants.

Response: Following your recommendation, we have carried out a one-week treatment with 30 μ M β -estrogen on four-week-old Col-0 plants, as shown below. Our results indicate that exogenous estrogen treatment has no discernible effect on the growth and development of Col-0 *Arabidopsis* plants. This is consistent with previous reports that a certain concentration of estrogen treatment does not affect plant growth.

The phenotype of 4-week-old pER8::*OBP3* plants after exposure to 30 μ M estrogen for one week. Bar = 4 cm.

Why do authors show phenotype for over-expression line in shoot and for mutant in the root? For consistency, author can show the phenotype of both root and shoot of mutant and over-expression line.

Response: In accordance with your recommendation, we have conducted a root growth analysis and recorded the fresh weight of the Col-0 and overexpression lines, which are presented in Supplementary Figure 3A. The results indicated that constitutive overexpression of *OBP3* inhibits root growth, resulting in a much shorter root and a lower fresh weight in the shoot. Additionally, we have performed the shoot and root growth analysis of mutants, which are illustrated in Figure 3C-3F. The shoot phenotype of *obp3* mutant is also demonstrated in figure 2, with a notable reduction in chlorophyll and Fe content in high pH soil. The shoot phenotype of the 35S::*OBP3* plant is also demonstrated in figure 2I.

Root phenotypic observation, root length analysis of wild-type Col-0 seedlings and genetically modified plants overexpressing *OBP3* at 9 days of age. Bar indicates 1cm. The error bar represents SD. Star indicates significant difference by students' *t*-test.

Over-expression of *bHLH100* in *obp3ilr3* double mutant background (35S::*bHLH100* (*obp3/ilr3*)) rescued the root growth phenotype under -Fe conditions. What was the observation of 35S::*bHLH100* (wt) phenotype?

Response: Following your advice, we have obtained the 35S::*bHLH100* in the Col-0 background. From the observation of the transgenic seedlings grown on the 1/2 MS medium, there is no significant growth difference between them and the wild type control at the seedling stage. These observations are in agreement with the previous report in the literature.

The observation of 7 days old Col-0 and the 35S::*bHLH100* (wt) seedling plants.

A single mutant of *obp3*, *ilr3*, and its double mutant *obp3ilr3* have the same primary root length phenotype. How does OBP3 and ILR3 interaction regulate the primary root growth and what is the significance of this interaction in root growth?

Response: Your question is very interesting and important. The observation that the phenotypes of *obp3* and *ilr3* mutant roots are similar to those of the wild type under sufficient iron conditions, but are defective under low iron conditions, suggests that these two regulatory genes play a specific role in the iron deficiency signalling pathway. Furthermore, the results of this phenotype are closely related to their relatively high expression and protein levels under iron deficiency conditions. In addition, we have included a discussion of this issue in the revised manuscript, as detailed in lines 440-443.

Minor comments:

Line 64: Check the typo, it should be bHLH100/ bHLH101

Response: Changed.

Line 66-68 It should be "subgroup Ib genes responsible for "activation of" iron uptake genes" and not" subgroup Ib genes responsible for iron uptake.

Response: Changed.

Line 130: Here authors have mentioned a transformation-based approach for Y1H. But in methodology, mating-based yeast is mentioned. Please clarify. If mating-based approach have used, please add plate images of screen as well.

Response: We have used the transformation-based approach for Y1H. We have corrected this error in the manuscript.

Line 144: Grown should be replaced with transferred.

Response: Changed.

Line 148: OBP3 cDNA have used as prey, right? and the bHLH100 promoter as a bait. Please correct.

Response: Corrected.

Line 166: Yeast one hybrid will verify the interaction only, not regulation. Right? Please correct.

Response: Corrected.

Line 188: OBP3 mutant generated or confirmed? as it was ordered from the NASC. Please clarify.

Response: Corrected.

Line 322: Dexamethasone will not induce OBP3 expression in the OBP3-GR line, it will change the localization of OBP3 from the cytoplasm to the nucleus.

Response: Corrected.

Line 591-592: It is important to discuss how Fe uptake is affected because of this regulation.

Response: Per your advice, we have added the discuss the regulation of Fe uptake as below:

“The observation that the phenotypes of *obp3* and *ilr3* mutant roots are similar to those of the wild type under sufficient iron conditions, but are defective under low iron conditions, suggests that these two regulatory genes play a specific role in the iron deficiency signalling pathway. ILR3 and OBP3 proteins interaction potentially leads to the formation of a functional heterodimer, which accumulates in low iron conditions and facilitates uptake under iron deficiency.”

Referee #2:

The manuscript titled "Dof3.6/OBP3, a DOF transcription factor gene, modulates iron homeostasis in Arabidopsis" by Hu et al. offers valuable insights into the regulatory mechanisms underlying iron homeostasis in Arabidopsis. It notably underscores the role of OBP3 as a transcription factor for iron-related genes and its interaction with the established iron sensor, BTS E3 ligase protein. The discovery of a novel role for OBP3 within the iron deficiency pathway significantly advances our understanding of the intricate network governing iron homeostasis. To further enhance the detailed comprehension of OBP3's functions, addressing the following critical concerns is essential to provide robust support for the conclusions:

Response: We would like to express our gratitude for your interest in our research. In light of your

valuable suggestions, we have implemented suitable modifications to effectively address the raised inquiries. We sincerely hope that the revised version meets your expectations and addresses the concerns you have raised.

1- Assess the Regulation of OBP3 Protein by BTS: Conduct experiments to determine dynamic changes in OBP3 protein abundance in response to varying iron concentrations. This would reinforce the observation that the OBP3 protein level is regulated by BTS.

Response: In accordance with your recommendation, supplementary experiments were conducted to ascertain the dynamic alterations in OBP3 protein abundance in response to Fe deficiency with varying iron concentrations. The findings indicated that OBP3 protein abundance was markedly elevated under lower Fe medium, which is in accordance with the mRNA expression level data presented in Figure 1D.

7 days old proOBP3::OBP3-GFP transgenic seedling plants cultivated under various concentration of Fe medium. OBP3 protein abundance under various concentration of Fe medium (100 μ M Fe, 5 μ M Fe and 0.5 μ M Fe) were shown using anti-GFP antibody. ACTIN was used as the internal control.

2- Investigate the Presence of a BTS Interaction Domain (BID) in OBP3: Strengthen the evidence for the potential interaction between OBP3 and BTS by exploring the presence of a BTS Interaction Domain (BID) in OBP3. Additionally, perform pull-down assays or ubiquitination assays to provide further support for the direct regulation of OBP3 by BTS.

Response: A thorough examination of the amino acid sequence of the OBP3 protein has not yet yielded the evidence of a BTS Interaction Domain (BID). The truncation experiment of OBP3 protein and the point-by-point verification experiment of yeast two-hybrid assay revealed that the C-terminal of OBP3 protein (181-368 aa) may mediate the interaction between OBP3 protein and BTS. Furthermore, an *in vitro* pull-down experiment was conducted to provide additional evidence for the direct interaction between OBP3 and BTS. The results of this experiment, along with the revised supplementary figure 8, are presented below. These results collectively reinforce the reliability of the direct interaction between the two proteins.

Y2H experiment was conducted to investigate the interaction between OBP3 $_{\Delta C}$, OBP3 $_{\Delta N}$ and BTS protein. The bait (pGBKT7-OBP3 $_{\Delta C}$ /OBP3 $_{\Delta N}$) and prey (BTS cloned into pGADT7) pairs were transformed into yeast cells, and selected on SD-Ade/-Trp/-Leu/-His medium. The yeast is diluted over a concentration gradient and then spotted onto a plate.

Supplemental Figure 8: The pull-down assay was conducted to verify the direct interaction between BTS and OBP3. The GST-BTS fusion protein and His-OBP3 fusion protein were purified using glutathione beads (GE Healthcare) and Ni-NTA agarose (GE Healthcare). 2 µg samples of BTS-GST-bound glutathione beads were incubated with 2 µg His-OBP3 in binding buffer containing 20 mM Tris-HCl, pH7.5 and 200 mM NaCl at 4°C for 2 h. Wash the beads three times with washing buffers (20mM Tris-HCl, pH7.5 and 200mM NaCl). Protein eluted from beads was loaded onto 12% SDS-PAGE gel by boiling in a 2 × sampling buffer of 50 µL and eluted by anti-His (GenScript, A00186) or anti-GST (GenScript, A00865-100) for WB analysis.

3-Validate OBP3's Regulatory Role on Transcription Factors: The study highlights OBP3's regulatory role on iron deficiency-induced transcription factors, including bHLH38/39/100/101, through binding to gene promoter regions with ILR3 protein. To validate this claim, examine the mRNA levels of these transcription factors, not just bHLH100, in the *obp3* and *ilr3* double mutant. This experiment would elucidate whether OBP3 and ILR3 work synergistically to regulate the expression of these downstream targets.

Response: Per your advice, and in order to better explain the expression level of the 1b bHLH family transcription factors in the background of *obp3* single and *obp3ilr3* double mutant, we further determined the expression level. The results showed that the expression level of *bHLH38/39/100/101* decreased to different degrees under the single mutation background, and the expression level was even lower under the *obp3ilr3* double mutation background. The results are shown in detail in the revised supplementary figure 7.

qPCR analysis was conducted on *bHLH38*, *bHLH39*, *bHLH101* expression levels in Col-0, *obp3* single and *obp3ilr3* double mutant backgrounds. The error bar represents SD. Star indicates significant difference by students' *t*-test.

4- The pronounced phenotype exhibited by the *obp3* mutant lines, particularly under iron-limiting conditions, warrants specific attention/discussion. Of note, a recent publication in Nature Plants (2023) has delineated the interconnected impact of iron content on plant growth, emphasizing that this relationship operates through an independent pathway (Zheng, Leiyang, et al. "The SOD7/DPA4-GIF1 module coordinates organ growth and iron uptake in Arabidopsis." Nature Plants 9.8 (2023): 1318-1332). Moreover, BioGrid data implies a potential interaction between OBP3 and GRF3 (<https://thebiogrid.org/10019/summary/arabidopsis-thaliana/obp3.html>). Significantly, mutations in GRF3 are correlated with diminished leaf size, suggesting its involvement in leaf development. This prompts an inquiry into whether the observed growth defects in the *obp3* mutant lines in this study are a direct consequence of altered iron content or are mediated through an independent pathway.

Response: Thank you for your interesting and important question. After careful analysis the phenotype of the *OBP3* overexpression lines and its mutant, we conclude that the involvement of *OBP3* in the regulation of plant iron homeostasis can be inferred from the following results. First, the expression level of *OBP3* was up-regulated by iron deficiency, and the protein level of OBP3

responded to iron deficiency and was significantly enriched under low-iron conditions. Second, the growth retardation phenotype caused by overexpression of *OBP3* was due to iron overload. Third, the iron content of the *obp3* mutant decreased significantly and was sensitive to iron deficiency. Fourth, OBP3 and the widely reported iron deficiency signalling gene ILR3 directly interact and act synergistically. Fifth, at the post-translational level, OBP3 is regulated by the ubiquitinated protein BTS, which fine-tunes iron homeostasis. In summary, we get the conclusion that *OBP3* is involved in the regulation of iron homeostasis and growth and development in Arabidopsis.

In conclusion, while the study significantly advances our understanding of iron homeostasis in Arabidopsis, the proposed experiments are recommended for enhanced clarity and further support the claims. By addressing these concerns, the manuscript's impact will undoubtedly be elevated, emphasizing the pivotal role of OBP3 in regulating iron homeostasis in Arabidopsis and providing valuable avenues for future research directions.

Response: We would like to express our gratitude for your interest in our research. In light of your valuable suggestions, we have implemented suitable modifications to effectively address the raised inquiries. We sincerely hope that the revised version fulfills your expectations.

Referee #3:

In this manuscript, Xu et al. identified the DOF transcription factor OBP3 in a yeast one-hybrid screen using the promoter of the Arabidopsis bHLH100 transcription factor as a bait. bHLH100 is a member of the bHLH clade Ib, which is known to regulate the iron deficiency response together with three other members of the subgroup. This study explores OBP3's role in Arabidopsis's iron deficiency response using mutants and reveals that OBP3 interacts with ILR3 to regulate bHLH100 expression and iron uptake. At the post-transcriptional level, the E3 ligase BRUTUS (BTS) assists in the degradation of the OBP3 protein by the 26S proteasome to prevent excessive iron uptake and maintain iron homeostasis in plants. The findings of this manuscript are novel and of interest to researchers on plant iron signalling and iron homeostasis regulation, the logic is reasonable, and the experiments are well performed. Therefore, this manuscript should be suitable for publication in this journal. However, I have some minor commendations listed below on this manuscript that need to be addressed before acceptance.

Response: We would like to thank you for your recognition of the importance of our research. Your suggestions are invaluable and greatly contribute to the constructive nature of our research. The questions you raised have been addressed through the implementation of appropriate modifications. We sincerely hope that the revised version meets your expectations.

1. Whether it is known that the promoter of bHLH100 is 1541 bp long.

Response: 1541 bp is the full-length promoter of bHLH100 that we have selected from the genome.

2. In Figure 1C, please indicate the negative control in the legend of the figure.

Response: The GUS activity corresponding to each promoter region cultured in iron-sufficient medium served as the control for each group. This information has been included in the revised figure legend.

3. The difference in GUS activity between +Fe and -Fe is difficult to appreciate. To show quantitative differences in promoter activity, quantitative GUS analysis is usually conducted.

Response: Based on your suggestion, we quantitatively analyzed GUS activity before and after iron deficiency treatment, as shown below.

(Related to Figure 1C) The figure depicts the relative GUS activity in proOBP3::GUS roots under Fe-deficient or Fe-sufficient conditions for 48 hours. The GUS activity under Fe-sufficient conditions was assigned a value of 1. The error bar represents the standard deviation. The star indicates a statistically significant difference ($p < 0.05$) as determined by a student's *t*-test.

4. To achieve a comprehensive understanding of OBP3 activity within this clade of bHLH transcription factors, it is advisable to incorporate expression data for the two additional genes, bHLH38 and bHLH101, in the manuscript.

Response: Following your recommendation, we conducted an experiment to demonstrate the expression of the remaining two genes, *bHLH38* and *bHLH101*, in the mutant lines. The findings indicate that all 1b subgroup bHLH genes, including *bHLH38* and *bHLH101*, are downregulated

in the *obp3* mutant background. The results are presented as below and have been incorporated into the revised supplementary figure 2B.

In the background of *obp3-2* mutant, iron deficiency induced upregulation of *bHLH38*, *bHLH39*, *bHLH100*, and *bHLH101* gene expression. The error bar represents SD. Star indicates significant difference by students *t* test.

5. The authors should assert that the expression analysis suggests a potential role for FERRITINS in mitigating oxidative damage.

Response: Changed as suggested in lines 244-245 of the revised manuscript.

6. What do FRO2 and IRT1 refer to ?

Response: FRO2 refers to the FERRIC REDUCTION OXIDASE 2 gene, while IRT1 denotes the IRON-REGULATED TRANSPORTER 1 gene.

7. Lines 76 and 86 should be defined.

Response: Based on your suggestion, we have defined the subgroup 1b genes and the DOF DNA binding motif in the revised manuscript.

8. In Figure 6A, what is the sequence of the DNA fragment used for the EMSA experiments?

Response: The sequence of the DNA fragment utilized in the EMSA assay is provided in Supplementary Table 5.

9. The growth conditions, including light intensity, and temperatures, are incomplete.

Response: Based on your suggestion, we have further supplemented the plant growth conditions as shown below and see the methods section of the revised manuscript.

“All mutants are in Col-0 background. Seeds were sterilized and vernalized at 4°C for 4-days before germination at 20°C. The Fe-sufficient medium consisted of half-strength Murashige and Skoog media with 1.0% (w/v) sucrose and 0.9% (w/v) agar, with 50 µM Fe-EDTA at pH 5.7. For Fe-deficient media, the same half-strength MS medium (without Fe) supplemented with 0.5 µM Fe-EDTA at pH 5.7. *A. thaliana* plants were grown at a temperature of 20°C under long-day conditions, which consisted of a 16-hour day and an 8-hour night, with a relative humidity of 60%. Protoplasts were prepared from three-week-old seedlings with rosette leaves. *N. benthamiana* were grown in climate-controlled growth chambers at 26°C for 6 weeks with 16 hours light and 8 hours dark.”

Dear Peipei,

We have now received re-review reports from two referees, which I have included below. As you will see, you have addressed their concerns satisfactorily; however, I would like you to consider addressing the remaining point of reviewer #2 in the discussion section. Before I can finally accept the manuscript, there are some remaining editorial points which need to be addressed. In this regard would you please:

- address the discrepancy between author names (Lihua Wang in the ms vs. Lihuang Wang) in online submission system and manuscript,
- remove emails from the title page of all authors except for the corresponding author,
- consider adding ORCIDiDs for authors,
- acknowledge funding from Natural Science Foundation of Shanghai (Grant No. 22ZR1469500) in our online submission system,
- correct reference list so that 'et al.' is used after 10 author names; only use DOIs for preprints and datasets that have not yet been published,
- include a 'Disclosure and competing interests statement'
- remove the author credit section from the manuscript,
- correct figure callouts so that Figure 4C is referred to and Figure 4D is called out before Figure 5C,
- upload Appendix 1 as a PDF with a table of contents with page numbers on the title page; the correct nomenclature (and for manuscript callouts) should be Appendix Figure S1, etc., Appendix Table S1, etc.,
- include a Reagents and Tools table,
- remove the number of figures/tables, supp. info from the manuscript's title page,
- change the title of the Materials and Methods to 'Methods',
- define the annotated p values * as well as provide the exact p-values for the same in the legend of figure 1c, f, h; 6d, and supplementary figure 2b,
- provide the exact p values in the legends of figures 1e; 2a, d; 3d-f; 4b-d; 6d-e; 7b-c, e-f, supplementary figure(s) 3a-c; 4a-b; 5; 7,
- indicate the statistical test used for data analysis in the legends of figures 1c, f, h; 6e, supplementary figure(s) 2b,
- correct the mismatch between the annotated p values in the figure legend and the annotated p values in the figure file for figures 3b, h-i; 6e; 7b-c,
- define 'n' in the legends of figures 1f; 3b; 7b-c, supplementary figure(s) 3c; 4a; 7,
- describe the nature of entity for 'n' in the legends of figures 1e, h,
- define error bars in the legends of figures 1f, supplementary figure(s) 3c; a,
- define centre for error bars in the legends of figures 1c, e, h; 2a; 3b; 6d-e, and supplementary figure(s) 2b; 7,
- the correct units for the scale bar in figures 1b and 2c is μm , not μM (in the figure legend), and
- define the asterisk in the legend of figure 6c.

We include a synopsis of the paper (see <http://emboj.embopress.org/>). Please provide me with a general summary image, and a two sentence statement and 3-5 bullet points that capture the key findings of the paper.

I am looking forward to receiving your revised manuscript.

EMBO Press is an editorially independent publishing platform for the development of EMBO scientific publications.

Best wishes,

William

William Teale, PhD
Editor
The EMBO Journal
w.teale@embojournal.org

We realize that it is difficult to revise to a specific deadline. In the interest of protecting the conceptual advance provided by the work, we recommend a revision within 3 months (15th Dec 2024). Please discuss the revision progress ahead of this time with the editor if you require more time to complete the revisions. Use the link below to submit your revision:

Referee #2:

In this revised version, the authors have effectively elucidated the role of OBP3 in response to iron deficiency in Arabidopsis. Their findings clearly demonstrate that OBP3 is involved in regulating iron deficiency by modulating the transcription of target transcription factors, including ILR3. However, it would be beneficial to provide clearer evidence in Figure 8 showing that OBP3 is regulated by BTS via ubiquitination. Despite this, the role of BTS as an upstream regulator of OBP3 is well-supported. The revised manuscript and additional experiments significantly enhance our understanding of the iron homeostasis pathway, particularly in relation to the transcription factor OBP3 in Arabidopsis, offering valuable insights for researchers in the field.

Referee #3:

The author has done quite a lot of supplementary work to our previous questions, and has answered my questions well. The quality of the paper has been significantly improved. I highly recommend the publication of this manuscript in EMBO J.

All editorial and formatting issues were resolved by the authors.

Dear Dr. Xu,

I am pleased to inform you that your manuscript has been accepted for publication in the EMBO Journal.

Congratulations!

Yours sincerely,

William Teale

William Teale, PhD
Editor
The EMBO Journal
w.teale@embojournal.org
